# A Tale of Two Features: Stable Diffusion Complements DINO for Zero-Shot Semantic Correspondence

**Junyi Zhang**[1]      **Charles Herrmann**[2]      **Junhwa Hur**[2]      **Luisa F. Polanía**[2]
**Varun Jampani**[2]      **Deqing Sun**[2]      **Ming-Hsuan Yang**[2,3]

[1] Shanghai Jiao Tong University      [2] Google Research      [3] UC Merced

https://sd-complements-dino.github.io

## Abstract

Text-to-image diffusion models have made significant advances in generating and editing high-quality images. As a result, numerous approaches have explored the ability of diffusion model features to understand and process single images for downstream tasks, *e.g.*, classification, semantic segmentation, and stylization. However, significantly less is known about what these features reveal across multiple, different images and objects. In this work, we exploit Stable Diffusion (SD) features for semantic and dense correspondence and discover that with simple post-processing, SD features can perform quantitatively similar to SOTA representations. Interestingly, our analysis reveals that SD features have very different properties compared to existing representation learning features, such as the recently released DINOv2: while DINOv2 provides sparse but accurate matches, SD features provide high-quality spatial information but sometimes inaccurate semantic matches. We demonstrate that a simple fusion of the two features works surprisingly well, and a zero-shot evaluation via nearest neighbor search on the fused features provides a significant performance gain over state-of-the-art methods on benchmark datasets, *e.g.*, SPair-71k, PF-Pascal, and TSS. We also show that these correspondences enable high-quality object swapping without task-specific fine-tuning.

## 1   Introduction

Recent findings from text-to-image diffusion models [44, 48, 50] have attracted increasing attention to analyze and understand their internal representations for a single image. Due to the text-to-image diffusion model's ability to effectively connect a text prompt to the content in the images, they should be able to understand what an image contains and were recently shown in [11, 31] to be effective classifiers. Furthermore, since these models can generate specific scenes and have some degree of control over the layout and placement of objects [72], they should be able to localize objects and were shown in [68] to be effective for semantic segmentation. In addition, several other methods [12, 17, 27, 65] focus on accessing and editing features for downstream tasks such as stylization and image editing.

However, these approaches have almost exclusively examined the properties of text-to-image diffusion features on a *single* image; significantly less is known about how these features relate across *multiple, different* images and objects. Simply put, in Stable Diffusion, how similar are the features of a cat to the features of a dog? In this paper, we focus on understanding how features in different images relate to one another by examining Stable Diffusion (SD) features through the lens of semantic correspondences, a classical vision task that aims to connect similar pixels in two or more images.

37th Conference on Neural Information Processing Systems (NeurIPS 2023).

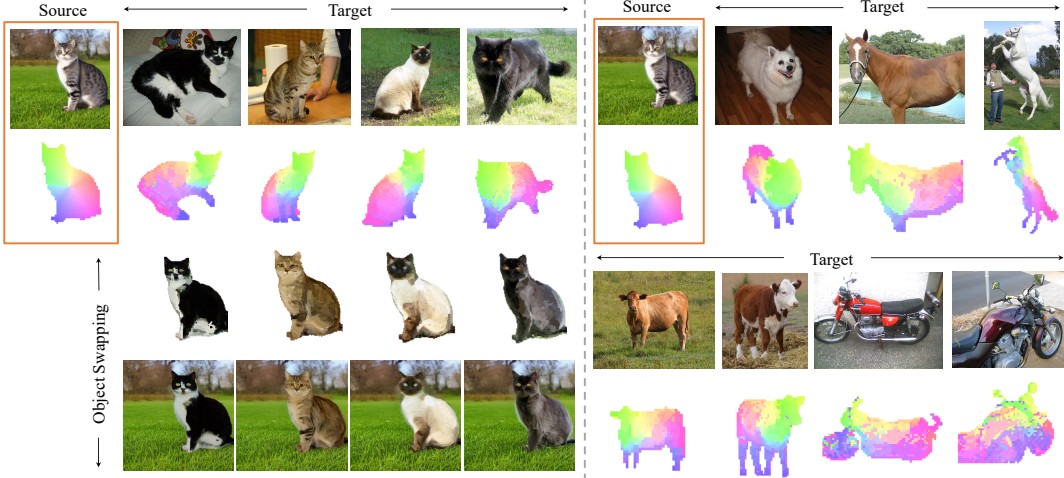

Figure 1: **Semantic correspondence with fused Stable Diffusion and DINO features.** On the *left*, we demonstrate the accuracy of our correspondences and demonstrate the instance swapping process. From top to bottom: Starting with pairs of images (source image in orange box), we fuse Stable Diffusion and DINO features to construct robust representations and build high-quality dense correspondence. This facilitates pixel-level instance swapping, and a subsequent stable-diffusion-based refinement process yields a plausible swapped instance. On the *right*, we demonstrate the robustness of our approach by matching dog, horses, cows, and even motorcycles to the cat in the source image. Our approach is capable of building reasonable correspondence even when the paired instances exhibit significant differences in categories, shapes, and poses.

For the correspondence task, we show that simply ensembling and reducing these features through basic techniques can lead to an effective representation that quantitatively does as well as other state-of-the-art representations for dense and semantic correspondences. However, a close analysis of the qualitative results leads to an interesting discovery. We observe that SD features have a strong sense of spatial layout and generate smooth correspondences (front and back of a bus are represented differently as shown in Fig. 3 left bottom), its pixel level matching between two objects can often be inaccurate (a bus front might match to another bus back). In fact, compared to existing features like DINOv1 [2], SD features appear to have very different strengths and weaknesses.

While prior work [2] showed that DINOv1 can be an effective dense visual descriptor for semantic correspondence and part estimation, to the best of our knowledge, this has not yet been thoroughly analyzed for DINOv2, which demonstrated improved downstream performance for multiple vision tasks but not correspondence. For correspondence, we show that DINOv2 does outperform DINOv1 but requires a different post-processing. DINOv2 generates sparse but accurate matches, which surprisingly, form a natural complement to the higher spatial information from SD features.

In this work, we propose a simple yet effective strategy for aligning and fusing these features. The fused representation utilizes the strengths of both feature types. To establish correspondence, we evaluate the zero-shot (no training for the correspondence task) performance of these features using nearest neighbors. Despite this very simple setup, our fused features outperform previous methods on the SPair-71k [36], PF-Pascal [20], and TSS [59] benchmark datasets. In addition, the fused representation facilitates other novel tasks such as instance swapping, where the objects in two images can be naturally swapped using estimated dense correspondence, while successfully maintaining its identity. The main contributions of this work are:

- We demonstrate the potential of the internal representation from a text-to-image generative model for semantic and dense correspondence.
- We analyze the characteristics of SD features, which produce spatially aware but inaccurate correspondences, and standard representation learning features, *i.e.*, DINOv2, which generate accurate but sparse correspondence and show that they complement each other.
- We design a simple strategy to align and ensemble SD and DINOv2 features and demonstrate that these features with a zero-shot evaluation (only nearest neighbors, no specialized training) can outperform many SOTA methods for semantic and dense correspondence. Experiments show that

the fused features boost the performance of the two and outperform previous methods significantly on the challenging SPair-71k dataset (+13%), as well as PF-Pascal (+15%) and TSS (+5%).

- We present an instance swapping application using high-quality correspondence from our methods.

## 2  Related work

**Feature descriptor for dense correspondence.** Deep models [15, 40, 45, 52, 66, 71] learn effective features for dense correspondence that can be robust to photometric and geometric changes such as rotation, scaling, or perspective transformation. However, most of the methods focus on the image matching task for outdoor scenes to account for rigid transformation. Amir *et al*. [2] demonstrate that features extracted from self-supervised Vision Transformer (DINOv1 [6]) serve as effective dense visual descriptors with localized semantic information by applying them to a variety of applications, *e.g*., (part) co-segmentation and semantic correspondences. Recently, DINOv2 [41] introduces a scaled-up version of DINOv1 [6] by increasing the model size and combining a large quantity of curated data. DINOv2 shows strong generalization to downstream tasks such as classification, segmentation, and monocular depth, but has not been extensively studied on correspondence tasks. Our work shows that DINOv2 also leads to significantly better correspondence results than DINOv1, but those DINO features in general produce sparse and noisy correspondence field.

**Semantic correspondence.** Semantic correspondence [33] aims to estimate dense correspondence between objects that belong to the same object class with different appearance, viewpoint, or non-rigid deformation. Conventional methods consist of three steps [64]: feature extraction, cost volume construction, and displacement field [60–63] or parameterized transformation regression [25, 28, 46, 47, 54]. Those methods, however, cannot effectively determine dense correspondence for complex articulated deformation due to their smooth displacement fields or locally varying parameterized (affine) transformations. Motivated by a classical congealing method [29], a couple of recent methods introduce to learn to align multiple objects in the same class by using pretrained DINOv1 feature [18, 39] or GAN-generated data [42]. The methods show the possibility that knowledge learned for different tasks can also be utilized for the dense correspondence task. However, those models are not effective in handling severe topological changes probably due to their strong rigidity assumption in the object alignment. In contrasts, we discover that the Stable Diffusion features for the image generation task exhibit great capability for the zero-shot dense correspondence task as well, where challenging articulate, non-rigid deformation exists.

**Diffusion models.** Diffusion probabilistic models (diffusion models) [55] and denoising diffusion probabilistic models [23, 38] have demonstrated state-of-the-art performance for image generation task [13, 37, 48, 50, 51, 57]. In addition, the diffusion models have been applied to numerous vision tasks, *e.g*., image segmentation [3, 4, 8, 26, 58, 67], object detection [7], and monocular depth estimation [14, 53]. Recently, much attention has been made to analyze what pre-trained diffusion models understand and extract its knowledge for single-image tasks, *e.g*., panoptic segmentation [68], semantic segmentation and depth [73]. In this paper, we unveil the potential of diffusion features for zero-shot dense correspondence between images.

## 3  Semantic correspondences via Stable Diffusion and Vision Transformer

We first examine properties of Stable Diffusion (SD) features for semantic correspondence, then study the complementary nature between DINO and SD features, and finally introduce a simple and effective fusion strategy to leverage the strengths of both features.

### 3.1  Are Stable Diffusion features good for semantic correspondence?

Stable Diffusion (SD) [48] demonstrates a remarkable ability to synthesize high-quality images conditioned on a text input, suggesting that it has a powerful internal representation for images and can capture both an image's content and layout. Several recent works [4, 68, 73] have used the pre-trained SD features for single-image tasks, such as semantic segmentation and depth perception. As features play an important role for visual correspondence, we are interested in investigating whether SD features can help establish semantic correspondences between images.

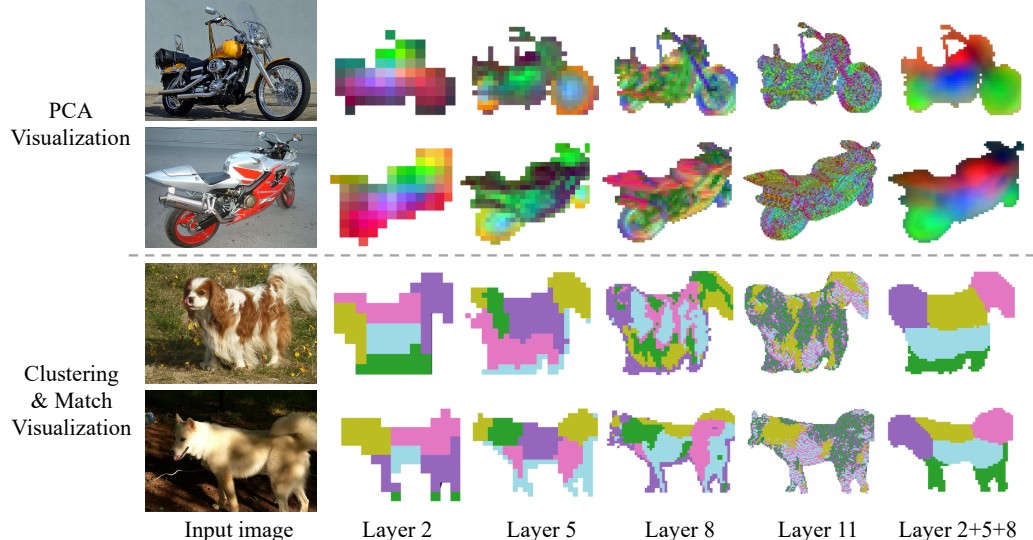

Figure 2: **Analysis of features from different decoder layers in SD.** *Top*: Visualization of PCA-computed features from early (layer 2), intermediate (layers 5 and 8) and final (layer 11) layers. The first three components of PCA, computed across a pair of segmented instances, serve as color channels. Early layers focus more on semantics, while later layers concentrate on textures. *Bottom*: K-Means clustering of these features. K-Means clusters are computed for each image individually, followed by an application of the Hungarian method to find the optimal match between clusters. The color in each column represents a pair of matched clusters.

Next, we briefly explain how we extract SD features. The architecture of Stable Diffusion consists of three parts: an encoder $\mathcal{E}$, a decoder $\mathcal{D}$ that is derived from VQGAN [16] and facilitates the conversion between the pixel and latent spaces, and a denoising U-Net $\mathcal{U}$ that operates in the latent space. We first project an input image $x_0$ into the latent space via the encoder $\mathcal{E}$ to produce a latent code $z_0$. Next, we add a Gaussian noise $\epsilon$ to the latent code according to a pre-defined time step $t$. Then, taking the latent code $z_t$ at the time step $t$ with the text embedding $C$ as inputs, we extract the features $\mathcal{F}_{\text{SD}}$ from the denoising U-Net. The entire process can be formally represented as follows:

$$z_0 = \mathcal{E}(x_0), \quad z_t = \sqrt{\bar{\alpha}_t}z_0 + \sqrt{1 - \bar{\alpha}_t}\epsilon, \quad \mathcal{F}_{\text{SD}} = \mathcal{U}(z_t, t, C), \tag{1}$$

where the coefficient $\bar{\alpha}_t$ controls the noise schedule, as defined in [23]. For the text embedding $C$, we follow [68] to use an implicit captioner, which empirically is superior to explicit text prompts .

While prior work [65] has reported that the intermediate U-Net layers have more semantic information for image-to-image translation task, it is unclear whether these features are suitable for semantic correspondence. To examine their properties, we first perform principal component analysis (PCA) on the features at various layers of the U-Net decoder. As shown in the top of Fig. 2, early layers tend to focus more on semantics and structures (*e.g.* wheels and seats) but at a coarse level; the later layers tends to contain detailed information about texture and appearance. Furthermore, we apply K-Means clustering ($k = 5$) on the features of paired instances and visually examine whether the clustered features contain consistent semantic information across intra-class examples. As shown in the bottom of Fig. 2, different parts of the objects are clustered and matched across the instance pairs. Both PCA and K-means analysis show that earlier layer features capture coarse yet consistent semantics, while later layers focus on low-level textural details. These observations motivate us to combine the features at different levels (2+5+8) to capture both semantic and local details, through concatenation.

A simple concatenation, however, results in an unnecessarily high-dimensional feature ($1280 + 960 + 480 = 2720$). To reduce the high dimension, we compute PCA across the pair of images for each feature layer, and then aggregate them together to the same resolution. Specifically, we first extract the $i^{\text{th}}$ layer's features for source and target images, $f_i^s$ and $f_i^t$. Next, we concatenate each layer's source feature and target feature and compute PCA together:

$$\tilde{f}_i^s, \tilde{f}_i^t = \text{PCA}(f_i^s \| f_i^t), \quad i \in \{2, 5, 8\}.$$

Then we gather each layer's dimension-reduced features $\tilde{f}_i^s$ and $\tilde{f}_i^t$, and upsample them to the same resolution to form the final SD feature $\tilde{f}^s$ and $\tilde{f}^t$. As shown in the last column of Fig. 2, the

Table 1: **Evaluation of correspondence on SPair-71k subsets.** We sample 20 pairs for each category and report the PCK@0.10 score for different settings. We underline the best performances achieved by each model. S/8: A small model with a patch size of 8, B/14: Base model with a patch size of 14.

| Model | Layer 11↑ | | | | Layer 9↑ | | | |
|---|---|---|---|---|---|---|---|---|
| | Token | Key | Query | Value | Token | Key | Query | Value |
| DINOv1-ViT-S/8 | 28.8 | 30.4 | 26.9 | 25.8 | 30.9 | 31.4 | 29.9 | 27.7 |
| DINOv2-ViT-S/14 | 52.7 | 30.3 | 30.6 | 47.1 | 45.5 | 12.7 | 13.2 | 40.6 |
| DINOv2-ViT-B/14 | 55.7 | 42.6 | 40.7 | 53.4 | 50.8 | 25.2 | 25.3 | 46.0 |

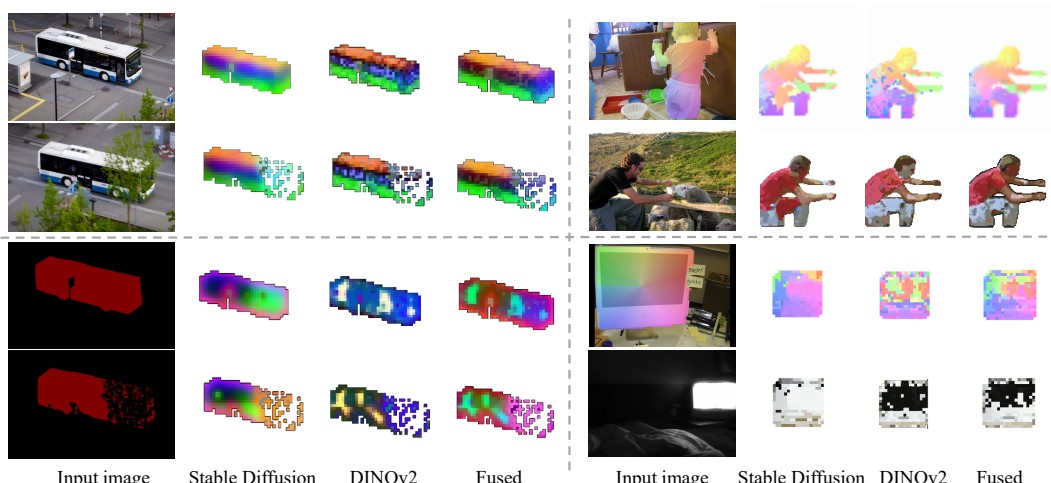

Figure 3: **Analysis of different features for correspondence.** We present visualization of PCA for the inputs from DAVIS [43] (left) and dense correspondence for SPair-71k [36] (right). The figures show the performance of SD and DINO features under different inputs: identical instance (top left), pure object masks (bottom left), challenging inputs requiring semantic understanding (right top) and spatial information (right bottom). Please refer to Supplemental B.1 for more results.

ensembled features strike a balance between the two different properties and focus on both semantics and textures at a sufficient resolution.

In addition, we find that the specific location within the decoder layer to extract the feature also plays a important role. U-Net's decoder layer inputs the feature of the previous decoder layer as well as the skip-connected encoder layer feature, and outputs the decoder feature after passing through a sequence of layers: a convolutional layer, a self-attention layer, and a cross-attention layer. We empirically find that using only the decoder feature achieves better performance than combining it with the encoder feature, and also outperforms the features of a single sub-layer (e.g., convolutional layer, self-attention layer). Please refer to appendix for more details.

One natural question is whether SD features provide useful and complementary semantic correspondences with respect to more widely studied discriminative features. Next, we discuss the semantic correspondence properties of the widely used DINOv1 [6] features in conjunction with the successor DINOv2 [41] features.

## 3.2 Evolution of DINO features for semantic correspondence

Caron *et al*. [6] show that self-supervised ViT features (DINO), "contain explicit information for semantic segmentation and are excellent k-NN classifiers." Amir *et al*. [2] further demonstrate that DINO features have several interesting properties that allow zero-shot semantic correspondence across images. Most recently, Oquab *et al*. [41] introduce DINOv2 by scaling up their training data and the model size and observe significant performance boost on numerous single-image tasks, such as segmentation and depth. In this work, we examine whether DINOv2 can significantly improve semantic correspondence across images.

We test different internal features from DINOv2 on the standard correspondence benchmark SPair-71k [36] (please refer to Sec. 4 for details). As shown in Tab. 1 PCK results (higher the better),

Table 2: **Smoothness (first-order difference) of the valid estimated flow field on the TSS dataset.** We report the results of three techniques and include the ground truth as a reference (lower is smoother).

| Method | FG3DCar↓ | JODS↓ | Pascal↓ | Avg.↓ |
|---|---|---|---|---|
| DINOv2-ViT-B/14 | 6.99 | 10.09 | 15.14 | 10.15 |
| Stable Diffusion | 3.48 | 7.87 | 8.44 | 5.90 |
| **Fuse-ViT-B/14** | 3.52 | 7.55 | 8.75 | 5.96 |
| Ground Truth | 2.22 | 5.20 | 4.06 | 3.40 |

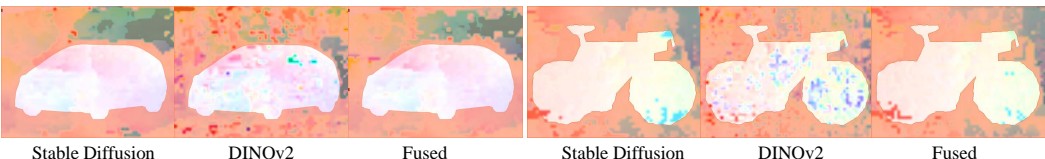

| Stable Diffusion | DINOv2 | Fused | Stable Diffusion | DINOv2 | Fused |
|---|---|---|---|---|---|

Figure 4: **Semantic flow maps** using different features. White mask indicates valid pixels and orange mask separates the background flow. SD features yield smoother flow fields versus DINOv2's isolated outliers.

DINOv2 features achieve a significant improvement over DINOv1. However, we observe that the best performance is achieved by the "token" facet from the last (11th) layer of DINOv2, while Amir et al. [2] find that, for DINOv1, the best performance is achieved by the "key" facet in earlier layers. We refer to the token features from layer 11 of DINOv2 as $\mathcal{F}_{\text{DINO}}$ or DINO.

### 3.3 Comparing the strengths and weaknesses of SD and DINO features

We now discuss the properties and potential complimentary nature of SD and DINO features. Aside from the PCA visualization above, we compute dense correspondences between image pairs using both features via nearest neighbor search and examine their behaviors under different input conditions.

**Correspondence for the same object instance.** For simpler cases where instances are the same in paired images (Fig. 3 top left), both SD and DINO features perform well. However, their performance differs on textureless examples. When giving an object mask as input (Fig. 3 bottom left), we observe that DINOv2 features cannot establish valid correspondence for this textureless example, while SD features work robustly as they have a strong sense of spatial layout.

**Correspondence between different object instances.** On examples with intra-class variation, both SD and DINO features perform well although in different regions. DINO yields more accurate matches (*e.g.*, in the thigh region of Fig. 3 top right case), but the correspondence field tends to be noisy, as demonstrated in both cases on the right side of Fig. 3. In contrast, SD excels at constructing smoother correspondence, as shown in both cases on the right side of Fig. 3, and provides crucial spatial information, notably in the bottom right case.

**Spatial coherence of the correspondence.** To analyze the spatial coherence of the estimated correspondence, we can extract the semantic flow, *i.e.*, the 2D motion vector for every pixel from the source image to the corresponding pixel in the target image. We then compute the smoothness, *i.e.* first-order difference, of the estimated semantic flow on the TSS dataset [59], as shown in Tab. 2. This analysis reveals the inherent smoothness in the flow maps generated by different methods. The Stable Diffusion features yield smoother results compared to DINOv2-ViT-B/14, suggesting that Stable Diffusion features exhibit better spatial understanding than the DINO features.

**Discussion.** Tab. 2 evaluates the spatial smoothness of the flow fields estimated by different methods on the TSS dataset. The features of Stable Diffusion lead to smoother dense correspondence than DINO features. Fig. 4 visually shows that the correspondence by DINO features contains moderate amounts of isolated outliers, while that by Stable Diffusion is more spatially coherent.

**Summary.** The studies above reveal the complementary nature of DINO and Stable Diffusion features. DINO features can capture high-level semantic information and excel at obtaining sparse but accurate matches. In contrast, Stable Diffusion features focus on low-level spatial information and ensure the spatial coherence of the correspondence, particularly in the absence of strong texture signals. The complementary properties of DINOv2 and diffusion features offer promising potential for enhancing the performance of semantic correspondence tasks. A natural question arises:

## 3.4 How to fuse SD and DINO features?

Based on the discussions above, we propose a simple yet effective fusion strategy to exploit SD and DINO features. The core idea is to independently normalize both features to align their scales and distributions, and then concatenate them together:

$$\mathcal{F}_{\text{FUSE}} = (\alpha||\mathcal{F}_{\text{SD}}||_2, \ (1-\alpha)||\mathcal{F}_{\text{DINO}}||_2) \tag{2}$$

where $|| \cdot ||_2$ denotes the L2 normalization, $\alpha$ is a hyperparameter that controls the relative weight of the two features. An important aspect of this fusion approach is the selection of the weight $\alpha$. Empirically, we find that $\alpha = 0.5$ offers a good balance between the two features, effectively leveraging their complementary strengths. See the supplementary material for the ablation studies.

Observing the surprising effectiveness of our simple fusion strategy in Fig. 3, we see notable improvements in challenging cases from the SPair-71k dataset. The combined features noticeably outperforms either features alone in all the cases. It not only demonstrates enhanced precision and reduced noise in correspondences but also retains the inherently smoother transitions and spatial information unique to the SD feature. Furthermore, as shown both quantitatively in Tab. 2 and qualitatively in Fig. 4, the performance of the fused features aligns remarkably closely with that of the SD features in retaining smoother transitions and spatial information.

In the later section, we further show the remarkable effectiveness of our simple fusion strategy through extensive comparison with existing methods on various public benchmark datasets.

## 4 Experiments and analysis

We first provide in-depth analyses of both sparse and dense correspondence tasks on public benchmark datasets. Then as a novel application of our approach, we present a simple, efficacious instance swapping method between paired images, utilizing dense correspondence from our method.

**Evaluation of features for correspondence.** We evaluate the extracted features in two settings. First, in a *zero-shot* setting, we follow [2] by searching the nearest neighbors directly on the feature maps of pair images. Second, for datasets with a training split, we add a bottleneck layer [22, 68] on top of the extracted features and finetune it in a *supervised* manner. It is guided by the same objective as in [35]: a CLIP-style symmetric cross entropy loss with respect to corresponding keypoints.

**Implementation details.** We employ the Stable Diffusion v1-5 model as our feature extractor, with the timestep for the diffusion model to be $t = 100$ by default. We use the input resolution $960 \times 960$ for the diffusion model and $840$ for the DINO model, which results in a feature map with a 60 resolution. For PCA, we adopt a nearly optimal approximation SVD algorithm [19] for better efficiency and reduce the feature dimension to 256 when fusing with DINO-ViT-B/14. When visualizing dense correspondence, we apply a segmentation mask obtained from [68] as a preprocessing step. All experiments are conducted on a single NVIDIA RTX3090 GPU.

**Datasets.** For the sparse correspondence task, we evaluate on two standard datasets, SPair-71k [36] and PF-Pascal [20], which consists of 70 958 and 1341 image pairs respectively, sampled from 18 and 20 categories. We evaluate the dense correspondence on TSS [59], the only dataset that provides dense correspondence annotations on 400 image pairs derived from FG3DCAR [32], JODS [49], and PASCAL [21] datasets.

**Metrics.** To evaluate the correspondence accuracy, we use the standard Percentage of Correct Keypoints (PCK) [70] metric with a threshold of $\kappa \cdot \max(h, w)$ where $\kappa$ is a positive integer. and $(h, w)$ denotes the dimensions of the bounding box of an instance in SPair-71k, and the dimensions of the images in PF-Pascal and TSS respectively, by following the same protocol in prior work [18, 39, 42].

### 4.1 Sparse correspondence

Tab. 3 provides our *zero-shot* and *supervised* evaluation on the SPair-71k dataset. Our zero-shot method (Fuse-ViT-B/14) significantly outperforms all methods including unsupervised methods as well as even previous supervised methods by a large margin, achieving a leading average PCK score of **64.0**. We also observe that our fusion strategy significantly improves the performance of the DINOv2 baseline (DINOv2-ViT-B/14), resulting in a PCK score improvement of **8.4p** (from 55.6 to 64.0). The Stable Diffusion features, not only being competitive to DINOv2 on the average PCK metric,

Table 3: **Evaluation on SPair-71k.** Per-class and average PCK@0.10 on test split. The methods are categorized into four types: strong supervised (S), GAN supervised (G), unsupervised with task-specific design ($U^T$), and unsupervised with only nearest neighboring ($U^N$). ∗: fine-tuned backbone. †: a trained bottleneck layer is applied on top of the features. We report *per image* PCK result for the (S) methods and *per point* result for other methods. The highest PCK among *supervised methods* and *all other methods* are highlighted in **bold**, while the second highest are underlined. Our NN-based method surpasses all previous unsupervised methods significantly.

| | Method | Aero | Bike | Bird | Boat | Bottle | Bus | Car | Cat | Chair | Cow | Dog | Horse | Motor | Person | Plant | Sheep | Train | TV | All |
|---|---|---|---|---|---|---|---|---|---|---|---|---|---|---|---|---|---|---|---|---|
| S | SCOT [34] | 34.9 | 20.7 | 63.8 | 21.1 | 43.5 | 27.3 | 21.3 | 63.1 | 20.0 | 42.9 | 42.5 | 31.1 | 29.8 | 35.0 | 27.7 | 24.4 | 48.4 | 40.8 | 35.6 |
| | CATs* [9] | 52.0 | 34.7 | 72.2 | 34.3 | 49.9 | 57.5 | 43.6 | 66.5 | 24.4 | 63.2 | 56.5 | 52.0 | 42.6 | 41.7 | 43.0 | 33.6 | 72.6 | 58.0 | 49.9 |
| | PMNC* [30] | 54.1 | 35.9 | 74.9 | 36.5 | 42.1 | 48.8 | 40.0 | 72.6 | 21.1 | 67.6 | 58.1 | 50.5 | 40.1 | 54.1 | 43.3 | 35.7 | 74.5 | 59.9 | 50.4 |
| | SCorrSAN* [24] | 57.1 | 40.3 | 78.3 | 38.1 | 51.8 | 57.8 | 47.1 | 67.9 | 25.2 | 71.3 | 63.9 | 49.3 | 45.3 | 49.8 | 48.8 | 40.3 | 77.7 | 69.7 | 55.3 |
| | CATs++* [10] | 60.6 | 46.9 | 82.5 | 41.6 | 56.8 | 64.9 | 50.4 | 72.8 | 29.2 | 75.8 | 65.4 | 62.5 | 50.9 | 56.1 | 54.8 | 48.2 | 80.9 | **74.9** | 59.9 |
| | DINOv2-ViT-B/14† | 80.4 | 60.2 | 88.1 | 59.5 | 54.9 | 82.0 | 73.5 | 89.1 | 53.3 | 85.5 | 73.6 | 73.8 | 65.2 | 72.3 | 43.6 | 65.6 | 91.4 | 60.3 | 69.9 |
| | Stable Diffusion† (**Ours**) | 75.6 | 60.3 | 87.3 | 41.5 | 50.8 | 68.4 | 77.2 | 81.4 | 44.3 | 79.4 | 62.8 | 67.7 | 64.9 | 71.6 | 57.8 | 53.3 | 89.2 | 65.1 | 66.3 |
| | Fuse-ViT-B/14† (**Ours**) | **81.2** | **66.9** | **91.6** | **61.4** | **57.4** | **85.3** | **83.1** | **90.8** | **54.5** | **88.5** | **75.1** | **80.2** | **71.9** | **77.9** | **60.7** | **68.9** | **92.4** | 65.8 | **74.6** |
| G | GANgealing [42] | - | 37.5 | - | - | - | - | - | 67.0 | - | - | 23.1 | - | - | - | - | - | - | - | 57.9 | - |
| $U^T$ | VGG+MLS [1] | 29.5 | 22.7 | 61.9 | 26.5 | 20.6 | 25.4 | 14.1 | 23.7 | 14.2 | 27.6 | 30.0 | 29.1 | 24.7 | 27.4 | 19.1 | 19.3 | 24.4 | 22.6 | 27.4 |
| | DINO+MLS [1, 5] | 49.7 | 20.9 | 63.9 | 19.1 | 32.5 | 27.6 | 22.4 | 48.9 | 14.0 | 36.9 | 39.0 | 30.1 | 21.7 | 41.1 | 17.1 | 18.1 | 35.9 | 21.4 | 31.1 |
| | NeuCongeal [39] | - | 29.1 | - | - | - | - | - | 53.3 | - | - | 35.2 | - | - | - | - | - | - | - | - | - |
| | ASIC [18] | 57.9 | 25.2 | 68.1 | 24.7 | 35.4 | 28.4 | 30.9 | 54.8 | 21.6 | 45.0 | 47.2 | 39.9 | 26.2 | 48.8 | 14.5 | 24.5 | 49.0 | 24.6 | 36.9 |
| $U^N$ | DINOv1-ViT-S/8 [2] | 57.2 | 24.1 | 67.4 | 24.5 | 26.8 | 29.0 | 27.1 | 52.1 | 15.7 | 42.4 | 43.3 | 30.1 | 23.2 | 40.7 | 16.6 | 24.1 | 31.0 | 24.9 | 33.3 |
| | DINOv2-ViT-B/14 | 72.7 | 62.0 | 85.2 | **41.3** | 40.4 | 52.3 | 51.5 | 71.1 | 36.2 | 67.1 | 64.6 | 67.6 | 61.0 | 68.2 | 30.7 | 62.0 | 54.3 | 24.2 | 55.6 |
| | Stable Diffusion (**Ours**) | 63.1 | 55.6 | 80.2 | 33.8 | 44.9 | 49.3 | 47.8 | 74.4 | 38.4 | 70.8 | 53.7 | 61.1 | 54.4 | 55.0 | 54.8 | 53.5 | 65.0 | 53.3 | 57.2 |
| | Fuse-ViT-B/14 (**Ours**) | **73.0** | **64.1** | **86.4** | 40.7 | **52.9** | **55.0** | **53.8** | **78.6** | **45.5** | **77.3** | **64.7** | **69.7** | **63.3** | **69.2** | **58.4** | **67.6** | **66.2** | **53.5** | **64.0** |

but also notably excel in certain categories where spatial information is critical (*e.g.*, "potted plant", "train", and "TV"). The same observation holds for supervised setting, notably the improvement in PCK score of **14.7p** (from 59.9 to 74.6) compared to previous method.

In Tab. 4, we further validate our approach on the PF-Pascal dataset, which is a less challenging dataset that presents low variations on appearance, pose, or shape between instance pairs. Our method consistently outperforms all unsupervised methods, achieving the highest average PCK scores across all thresholds. Our fusion approach (Fuse-ViT-B/14), again, substantially improves the performance over the DINOv2 baseline (DINOv2-VIT-B/14), highlighting the effectiveness of our fusion strategy that benefits from both features.

## 4.2 Dense correspondence

Tab. 4 further provides the dense correspondence evaluation on the TSS dataset [59]. Our fusion approach (Fuse-ViT-B/14) outperforms all unsupervised nearest-neighbor-search-based methods ($U^N$) on the TSS dataset. The result again confirms the effectiveness of our fusion strategy, demonstrating a substantial gain of **7.7p** over the DINOv2-ViT-B/14 baseline. Note that the TSS dataset contains less challenging examples with low variations on appearance, viewpoint, and deformation (*e.g.* car, bus, train, plane, bicycle, *etc*.), where having a strong spatial smoothness prior can be an advantage for achieving good metrics on the benchmark.

## 4.3 Instance swapping

Given a pair of images with instances of the same or similar categories, instance swapping is defined as the task of transposing an instance from a target image (target instance) onto the instance(s) in a source image (source instance), while preserving the identity of the target instance and the pose of the source instance. This creates a novel image where the target instance appears naturally integrated into the source environment.

By leveraging the high-quality dense correspondence obtained by our method, we can achieve plausible instance swapping through a straightforward process: 1) Initially, we upsample the low-resolution feature map to match the size of the target image; 2) Based on the upsampled feature map, we perform pixel-level swapping on the segmented instances according to the nearest neighbor patch, yielding a preliminary swapped image; 3) To enhance the quality of the swapped image, we refine the preliminary result with a stable-diffusion-based image-to-image translation process [65].

Table 4: **Evaluation on PF-Pascal and TSS.** The highest PCK among *nearest-neighboring based unsupervised* are highlighted in **bold**, while the second highest are underlined. Our fusion results result in a large improvement over the DINO baselines, and are comparable to other task-specific methods. S: Supervised methods, $U^T$: Task-specific unsupervised methods, $U^N$: Nearest-neighboring based unsupervised methods. $*$: fine-tuned backbone. $\dagger$: a trained bottleneck layer is applied on top of the features.

| | | PF-Pascal, PCK@$\kappa$ | | | TSS, PCK@0.05 | | | |
|---|---|---|---|---|---|---|---|---|
| | Method | 0.05 | 0.10 | 0.15 | FG3DCar | JODS | Pascal | Avg. |
| S | SCOT [34] | 63.1 | 85.4 | 92.7 | 95.3 | 81.3 | 57.7 | 78.1 |
| | CATs$^*$ [9] | 76.8 | 92.7 | 96.5 | 92.1 | 78.9 | 64.2 | 78.4 |
| | PWarpC-CATs$^*$ [64] | 79.8 | 92.6 | 96.4 | 95.5 | 85.0 | 85.5 | 88.7 |
| | CATs++$^*$ [10] | 84.9 | 93.8 | 96.8 | - | - | - | - |
| | DINOv2-ViT-B/14$^\dagger$ | 74.2 | 90.8 | 95.4 | - | - | - | - |
| | Stable Diffusion$^\dagger$ (**Ours**) | 77.4 | 89.7 | 93.9 | - | - | - | - |
| | Fuse-ViT-B/14$^\dagger$ (**Ours**) | 80.9 | 93.6 | 96.9 | - | - | - | - |
| $U^T$ | CNNGeo [46] | 41.0 | 69.5 | 80.4 | 90.1 | 76.4 | 56.3 | 74.4 |
| | PARN [25] | - | - | - | 89.5 | 75.9 | 71.2 | 78.8 |
| | GLU-Net [60] | 42.2 | 69.1 | 83.1 | 93.2 | 73.3 | 71.1 | 79.2 |
| | Semantic-GLU-Net [63] | 48.3 | 72.5 | 85.1 | 95.3 | 82.2 | 78.2 | 85.2 |
| $U^N$ | DINOv1-ViT-S/8 [2] | 41.5 | 62.4 | 72.5 | 64.7 | 51.2 | 36.7 | 53.3 |
| | DINOv2-ViT-B/14 | 56.2 | 77.3 | 83.3 | 82.8 | **73.9** | 53.9 | 72.0 |
| | Stable Diffusion (**Ours**) | 61.0 | 80.3 | 86.1 | 93.9 | 69.4 | 57.7 | 77.7 |
| | Fuse-ViT-B/14 (**Ours**) | **73.0** | **86.1** | **91.1** | **94.3** | 73.2 | **60.9** | **79.7** |

Source     Target        Stable Diffusion     DINOv2     Fused

Figure 5: **Qualitative comparison of instance swapping with different features.** SD features deliver smoother swapped results, DINOv2 reveals greater details, and the fused approach takes the strengths of both. Notably, the fused features generate more faithful results to the reference image, as highlighted by the preserved stripes on the cat instance in the top-right example. Please refer to Supplemental B.2 for more results.

To be more specific with the refinement process, we begin by inverting the swapped image to the initail noise with DDIM inversion [56], followed by a denoising DDIM sampling process where we extract the features from the diffusion model. Finally, merging this with a prompt that includes the instance category, we generate a refined image. This image exhibits a swapped instance with an appearance that is visually more coherent and plausible. Please refer to Supplemental A.8 and B.2 respectively for the quantitative and qualitative analysis of the refinement process.

Fig. 5 provides a visual comparison of instance swapping using different features. SD leads to smooth results, whereas DINOv2 accentuates fine details. The fusion of both methods showcases a balance between spatial smoothness and details. More examples are available in the supplementary material.

We also compare different features quantitatively on an 800-pair subset of [69] benchmark. The quantitative evaluation in Tab. 5 reveals consistent performance gains of the fused approach across all three metrics: FID score, quality score, and CLIP score. We follow the same metrics from [69]: FID score is measured using the CLIP features between the generated images and the COCO 2017 testing images, and CLIP score measures the cosine similarity between CLIP features extracted from the edited region of the source image and the target image.

These results show the strength of the dense correspondences with fused features in not only maintaining the instance's appearance but also generating high-quality images. The CLIP score is higher for

Table 5: **Quantitative comparison for instance swapping.** The best performance approach is in **bold**.

| Method | FID score(CLIP-based)↓ | Quality score↑ | CLIP score↑ |
|---|---|---|---|
| DINOv2-ViT-B/14 | 12.49 | 63.18 | 72.63 |
| Stable Diffusion | 13.72 | 61.38 | 71.48 |
| Fuse-ViT-B/14 | **12.47** | **64.84** | **73.21** |

Table 6: **Distribution of outcomes under different datasets and PCK levels.** Under most settings, one feature succeeds while the other fails in 20~30% of total cases (see row 2 & 3), which suggests that they have a substantial amount of non-redundant information.

| | SPair-71k, PCK@$\kappa$ | | | PF-Pascal, PCK@$\kappa$ | | |
|---|---|---|---|---|---|---|
| Cases | 0.15 | 0.10 | 0.05 | 0.15 | 0.10 | 0.05 |
| SD, DINO fails | 21.7 | 29.2 | 44.5 | 5.6 | 10.0 | 27.1 |
| SD fails, DINO correct | 15.7 | 15.8 | 14.2 | 8.3 | 9.7 | 12.0 |
| SD correct, DINO fails | 14.0 | 15.3 | 15.8 | 11.1 | 12.7 | 16.8 |
| SD, DINO correct | 48.6 | 39.7 | 25.5 | 75.0 | 67.6 | 44.2 |

the fused approach, indicating that the fused feature representation is more faithful to the reference image. Meanwhile, the higher quality scores and lower FID scores point to the superior perceptual quality and semantic coherence of the images generated by the fused approach.

### 4.4 Limitations

Although our approach shows promising correspondence performance, it does come with certain limitations. The relatively low resolution of the fused features impedes the construction of precise matches, which particularly required in dense correspondence tasks as in the TSS dataset. Detailed exploration of failure cases can be found in the supplementary material. Additionally, the integration of the Stable Diffusion model, while only requiring a single inference, considerably increases the cost of the correspondences computation in comparison to using DINO features only.

### 4.5 Quantitative analysis of feature complementary

We present additional quantitative analysis on the non-redundancy of SD and DINOv2 features in Tab. 6, which details the error distribution for these two features on SPair-71k and PF-Pascal benchmarks at three different PCK levels. In 20~30% of total cases under most settings, one feature succeeds where the other fails, suggesting that they have a substantial amount of non-redundant information. For a more detailed analysis on this non-redundancy, please refer to Supplemental A.10.

## 5 Discussion on feature behavior

The distinct behaviors observed in Stable Diffusion (SD) and DINOv2 features have spurred questions regarding the underlying causes. Though it is challenging to validate due to resource limitations, we suggest some possible explanations: **Training paradigms,** DINO's self-supervised learning approach could induce an invariance to spatial information because of its training augmentations. This contrasts with SD, which, being trained for text-to-image synthesis, inherently demands awareness of both global and local visual cues, possibly leading to its heightened spatial sensitivity. **Architectural differences,** DINOv2, built upon the ViT model, interprets images as sequences of patches. Even with positional encoding, it might not prioritize local structures as much. On the other hand, the convolution layers in SD's UNet may retain more details and enhance the retention of spatial information. In general, identifying the causes of these variances remains an intriguing topic and is a great direction for future exploration.

## 6 Conclusion

In this work, we explore the potential of leveraging internal representations from a text-to-image generative model, namely Stable Diffusion, for the semantic correspondence task. We reveal the complementary nature of the Stable Diffusion features and self-supervised DINO-ViT features, providing a deeper understanding of their individual strengths and how they can be effectively combined. Our proposed simple strategy of aligning and fusing these two types of features has empirically proven to significantly enhance performance on the challenging SPair-71k dataset and outperform existing methods, suggesting the benefits of pursuing better features for visual correspondence.

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
