# A Tale of Two Features: Stable Diffusion Complements DINO for Zero-Shot Semantic Correspondence
## – Supplementary Material –

## Contents

## A   Additional analyses

To further understand the contributions of each component in our method as well as the impact of various design choices, we conduct a series of ablation studies on the SPair-71k dataset [7]. The quantitative results are reported in terms of PCK at different $\kappa$ thresholds, and we sample 20 pairs for each category.

### A.1   Effect of UNet layers

Table 1: **Ablation study on SPair-71k: number of UNet layer.** We report PCK@$\kappa$ ($\kappa = 0.01, 0.05, 0.10$) for each setting and both the Stable Diffusion and Fuse-ViT-B/14 methods. Default setting is underlined.

| Setting | | | Stable Diffusion | | | Fuse-ViT-B/14 | | |
|---|---|---|---|---|---|---|---|---|
| | | | 0.01 | 0.05 | 0.10 | 0.01 | 0.05 | 0.10 |
| 2 | 5 | 8 | 1.2 | 19.2 | 39.2 | 5.7 | 40.1 | 58.3 |
| 2 | 5 | 8 | 4.9 | 39.5 | 54.2 | 7.0 | 46.4 | 61.6 |
| 2 | 5 | 8 | 4.3 | 23.9 | 31.7 | 7.0 | 42.2 | 55.5 |
| 2 | 5 | 8 | 4.8 | 38.8 | 53.1 | 6.9 | 45.7 | 61.6 |
| 2 | 5 | 8 | 4.5 | 30.5 | 42.6 | 7.6 | 43.0 | 59.3 |
| 2 | 5 | 8 | 5.7 | 40.5 | 53.0 | 7.6 | 46.8 | 61.8 |
| 2 | 5 | 8 | 5.6 | 39.9 | 53.4 | 7.7 | 46.2 | 62.2 |

We analyze how features extracted at different layers in the U-Net architecture affect the accuracy, specifically at layers 2, 5, and 8, for the Stable Diffusion (SD) and Fuse-ViT-B/14 methods. The experiment results in Tab. 1 suggest that layer 5 alone provides substantial performance for both the Stable Diffusion and the fused features, while gathering all three layers further improves the overall performance for the fused features.

Table 2: **Ablation study on SPair-71k: location of UNet layer.** We report PCK@$\kappa$ ($\kappa = 0.01, 0.05, 0.10$) for each setting and both the Stable Diffusion and Fuse-ViT-B/14 methods. Default setting is underlined.

| Setting (with the layer 5) | Stable Diffusion | | | Fuse-ViT-B/14 | | |
|---|---|---|---|---|---|---|
| | 0.01 | 0.05 | 0.10 | 0.01 | 0.05 | 0.10 |
| Conv. features | 4.0 | 25.3 | 37.1 | 6.7 | 44.3 | 57.3 |
| Self-attn features | 4.1 | 23.1 | 34.0 | 6.1 | 45.0 | 56.7 |
| Encoder features | 3.2 | 19.8 | 27.7 | 5.3 | 39.1 | 53.5 |
| Encoder & Decoder features | 4.9 | 39.5 | 54.2 | 7.0 | 46.4 | 61.6 |
| Decoder features | **5.3** | **39.9** | **54.8** | **7.1** | **47.0** | **62.1** |

We also analyze how the exact location of feature extraction affects the performance. As shown in Tab. 2, we empirically find that using only the decoder feature achieves better performance than combining it with the skip-connected encoder feature, and also outperforms the features of a single sub-layer such convolutional or self-attention layer.

### A.2   Effect of dimensionality reduction

We assess the effects of different dimension reduction techniques, specifically the projection layer pre-trained on panoptic segmentation as presented in [12], as well as Principal Component Analysis (PCA) with varying projection dimensions. As shown in Tab. 3, we observe that the projection layer from [12], owing to its training on a different task, leads to a performance drop. On the other hand, both the stable diffusion features and the fused features exhibit robustness with respect to different projection dimensions in PCA.

### A.3   Effect of fusion strategy

We experiment with different fusion strategies for combining the feature maps from stable diffusion and DINO. The strategies include element-wise addition ($\mathcal{F}_{\text{FUSE}} = \mathcal{F}_{\text{SD}} + \mathcal{F}_{\text{DINO}}$, both with and

Table 3: **Ablation study on SPair-71k: Reduction.** We report PCK@$\kappa$ ($\kappa = 0.01, 0.05, 0.10$) for each setting and both the Stable Diffusion and Fuse-ViT-B/14 methods. Default setting is underlined.

| Setting | Stable Diffusion | | | Fuse-ViT-B/14 | | |
|---|---|---|---|---|---|---|
| | 0.01 | 0.05 | 0.10 | 0.01 | 0.05 | 0.10 |
| Projection layer [12] | 5.7 | 34.8 | 46.0 | 7.5 | 44.9 | 59.9 |
| PCA (dim = 384) | 5.7 | 39.5 | 53.0 | 7.7 | 46.3 | 61.8 |
| PCA (dim= 256) | 5.6 | 39.9 | 53.4 | 7.7 | 46.2 | 62.2 |
| PCA (dim = 128) | 5.4 | 39.9 | 52.8 | 7.4 | 45.7 | 61.9 |

Table 4: **Ablation study of Fuse-ViT-B/14 on SPair-71k: Fuse Strategy.** We report PCK@$\kappa$ ($\kappa = 0.01, 0.05, 0.10$) for each setting. Default setting is underlined.

| Setting | 0.01 | 0.05 | 0.10 |
|---|---|---|---|
| $\mathcal{F}_{\text{FUSE}} = \mathcal{F}_{\text{SD}} + \mathcal{F}_{\text{DINO}}$ | 6.8 | 43.6 | 57.7 |
| $\mathcal{F}_{\text{FUSE}} = \|\mathcal{F}_{\text{SD}}\|_2 + \|\mathcal{F}_{\text{DINO}}\|_2$ | 6.9 | 43.7 | 57.8 |
| $\mathcal{F}_{\text{FUSE}} = (\mathcal{F}_{\text{SD}}, \mathcal{F}_{\text{DINO}})$ | 6.7 | 43.6 | 57.9 |
| $\mathcal{F}_{\text{FUSE}} = (\|\mathcal{F}_{\text{SD}}\|_2, \|\mathcal{F}_{\text{DINO}}\|_2)$ | 7.7 | 46.2 | 62.2 |

without independent normalization), concatenation without normalization ($\mathcal{F}_{\text{FUSE}} = (\mathcal{F}_{\text{SD}}, \mathcal{F}_{\text{DINO}})$), the results demonstrate that our fusion strategy, only concatenation with normalization, stands out among the others, as shown in Tab. 4.

## A.4 Effect of captioner and timestep

Table 5: **The PCK performance on SPair-71k for both implicit and explicit captioner under different timesteps.** Best results between captioner are **bold**, best results among different timesteps are *italicized*.

| Method | Captioner | 0 | 50 | 100 | 150 | 200 |
|---|---|---|---|---|---|---|
| SD | Implicit | **54.93** | **55.67** | ***56.18*** | 55.11 | 55.11 |
| | Explicit | 53.58 | 55.63 | *55.90* | **55.45** | **55.15** |
| Fuse | Implicit | **63.25** | **63.10** | ***63.28*** | 62.46 | 62.50 |
| | Explicit | 62.20 | 62.50 | 62.61 | **62.72** | ***62.32*** |

We further analyze the influence of timesteps during feature extraction and the use of either implicit or explicit captioners. Tab. 5 reports the PCK@0.10 performance of Diffusion features and Fused features in the SPair-71k 20-samples subset, when implicit and explicit textural (specifically, "a photo of {object category}") inputs are given.

**Captioner.** Overall, there are only marginal differences. The explicit textual inputs help in earlier steps (200 steps), while implicit captioner helps in denoised images. We conjecture that this is due to the implicit captioner from ODISE [12] being trained with $t = 0$.

**Timestep.** Our method extracts features at the time step 100 of 1000. As shown in Tab. 5, changing the timestep for feature extraction yields only marginal variations in accuracy. The timestep 100 was determined to be optimal through a search on the validation set.

ODISE [12] finds that $t = 0$ yields optimal results. This would be the case for semantic segmentation where a denoised image with clear object boundaries is critical for the accuracy. However, for semantic correspondence where semantic information is also important, feature maps with more structural information at a little bit earlier denoising step may help better.

## A.5 Effect of model variants

**Stable Diffusion.** We conduct extensive evaluations on multiple variants of the SD model, varying both in architecture and training configurations. Specifically, alternative training settings such as SD-1-

Table 6: **Comparison of different variants of SD models on SPair-71k**, we use explicit captioner for fair comparison. The best results are **bold**.

| Method | PCK@0.10 | PCK@0.05 | PCK@0.01 |
|---|---|---|---|
| SD-tiny | 41.07 | 28.67 | 5.21 |
| SD-small | 51.05 | 38.33 | 6.28 |
| SD-1-3 | 55.30 | 42.72 | **7.72** |
| SD-1-5 | 55.90 | **42.76** | 7.01 |
| SD-2-1-base | 54.43 | 41.68 | 7.19 |
| SD-XL | **56.46** | 40.26 | 5.60 |
| DINOv2-vitb14 | 55.15 | 39.66 | 6.12 |
| Fuse-vitb-tiny | 56.96 | 42.35 | 7.27 |
| Fuse-vitb-small | 60.36 | 45.08 | 7.95 |
| Fuse-vitb-1-3 | **62.69** | **47.09** | 8.25 |
| Fuse-vitb-1-5 | 62.61 | 46.60 | **8.47** |
| Fuse-vitb-2-1-base | 62.22 | 46.10 | 8.40 |
| Fuse-vitb-XL | 61.31 | 45.00 | 7.99 |

3 and SD-2-1-base are assessed alongside the main SD model. Additionally, distilled architectures [6] like SD-tiny and SD-small, which respectively comprise 45% and 65% fewer parameters than the base model, are considered. Plus, we also test a larger version of SD, i.e., SD-XL [9], with $3\times$ larger UNet as SD-1-5. As reported in Tab. 6, the performance variations across these SD base models are relatively minor. Yet, when combined with DINOv2, even the distilled variants, despite their slight performance deficits in isolation, lead to notable improvements.

Table 7: Comparison of different variants of DINO models. The best results are **bold**.

| Method | PCK@0.10 | PCK@0.05 | PCK@0.01 |
|---|---|---|---|
| DINOv1-vitb16 | 33.17 | 19.93 | 2.43 |
| iBOT-vitb16 | 38.85 | 23.90 | 2.63 |
| DINOv2-vits14 | 53.28 | 37.20 | 5.86 |
| DINOv2-vitb14 | 55.15 | 39.66 | 6.12 |
| Stable Diffusion | 56.18 | 42.80 | 6.79 |
| Fuse-DINOv1-vitb16 | 51.79 | 37.50 | 5.34 |
| Fuse-iBOT-vitb16 | 55.11 | 38.99 | 4.85 |
| Fuse-DINOv2-vits14 | 61.34 | 46.57 | 7.84 |
| Fuse-DINOv2-vitb14 | 63.28 | 47.61 | 8.32 |

**DINO.** We also test different DINOv2 variants, including a smaller version, DINOv2-vits14, which has about 25% network parameters of the base DINOv2 model, and two variants with different training objectives and datasets, namely DINOv1 and iBOT. As shown in Tab. 7, DINOv2 small model, though with substantially less parameters, still delivers comparable results to the base model. This suggests that while capacity plays a role, the core techniques remain effective even with a significantly smaller model. However, for DINOv1 and iBOT, a zero-shot fusion with SD slightly decreases the overall performance. We hypothesize that this may be due to the relatively weak performance of DINOv1 and iBOT; if these features are strictly worse than SD features, they may only contribute noise to the zero-shot fusion results. A learned projection would enable fusion to ignore features that decrease the overall performance and at least perform similarly to SD by itself.

## A.6 Effect of keypoint annotation and input image resolution

We assess the implications of varying the resolution of both keypoint annotations and input images on the performance. An in-depth examination of the PCK performance on the SPair-71k 20-samples subset, considering multiple input resolutions and annotation resolutions, can be found in Tab. 8. Notably, similar to the observations from PWarpC's [10] Table 1, using different annotation resolutions only marginally affects the accuracy. On the other hand, as in CATs++ [4], we also observe that input image's resolution matters more, especially under stricter constraints (*e.g.*, PCK@0.05 and 0.01).

Table 8: **PCK performance of fused features on SPair-71k**, under different input image resolution and keypoint annotation resolution.

| Input Image Resolution | Annotation Resolution | 0.10 | 0.05 | 0.01 |
|---|---|---|---|---|
| | 840 | 63.28 | 47.61 | 8.32 |
| 960 (feat. map 60*60) | original | 62.80 | 47.36 | 8.11 |
| | 256 | 62.54 | 47.58 | 8.36 |
| | 840 | 61.66 | 40.73 | 4.58 |
| 512 (feat. map 32*32) | original | 61.58 | 40.43 | 4.28 |
| | 256 | 61.51 | 40.35 | 4.40 |

## A.7  Effect of fusion weight $\alpha$

As denoted in Eq. (2) in the main paper, our fusion feature is a weighted sum of two normalized features, Stable Diffusion (SD) and DINO, $\mathcal{F}_{\text{FUSE}} = (\alpha||\mathcal{F}_{\text{SD}}||_2, (1-\alpha)||\mathcal{F}_{\text{DINO}}||_2)$, where $\alpha$ is a hyper parameter that controls the balance of the contributions between the two features. We examine how different $\alpha$ values affect correspondence matching results via qualitative analysis as shown in Fig. 1.

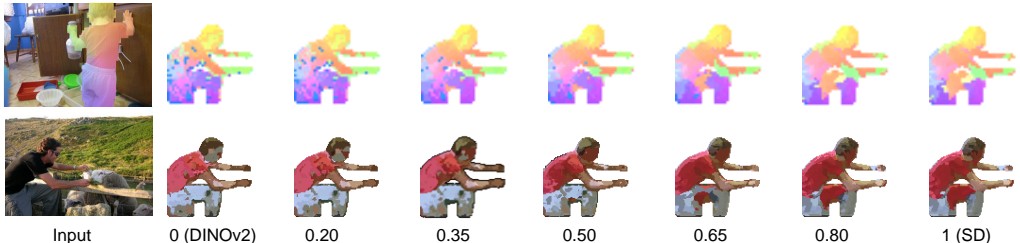

Figure 1: **Visualization of the dense correspondence across varying $\alpha$.** We visualize both the color map and the swapped image corresponding to the source image.

As observed in Fig. 1, as $\alpha$ shifts from 0 (entirely DINO features) to 1 (purely SD features), the noisy correspondence map gradually becomes smooth, but slightly less precise (*e.g.* the left thigh part of the person). Notably, $\alpha = 0.5$ strikes a beneficial balance, harnessing the complementary strengths of both features.

## A.8  Effect of refinement for instance swapping

In Sec. 4.3, we introduce a refinement step in our instance swapping process, aiming to elevate the quality and the realistic of the swapped instances. In this section, we provide a quantitative assessment of this refinement process.

Table 9: **Quantitative comparison of different methods on instance swapping.** We report the FID score, quality score, and CLIP score for three methods with and without refinement. The best results are **bold**.

| Method | FID score (CLIP-based)↓ | Quality score↑ | CLIP score↑ |
|---|---|---|---|
| DINOv2 (w/o refinement) | 11.28 | 60.43 | 67.72 |
| SD (w/o refinement) | 12.01 | 58.32 | 65.99 |
| Fused (w/o refinement) | **10.93** | **62.03** | **68.25** |
| DINOv2 (w/ refinement) | 12.49 | 63.18 | 72.63 |
| SD (w/ refinement) | 13.72 | 61.38 | 71.48 |
| Fused (w/ refinement) | **12.47** | **64.84** | **73.21** |

Tab. 9 depicts the influence of the refinement step on the resulting Quality scores and CLIP similarity scores. Remarkably, we observe a significant increase in these scores after applying the refinement. This improvement is particularly pronounced for the CLIP similarity scores, validating the intended benefit of this additional step.

However, an unexpected outcome of the refinement process is observed in the Fréchet Inception Distance (FID) scores, which tend to increase after refinement across all methods. This suggests that while the refinement step improves certain metrics, it may also introduce certain artifacts that amplify the discrepancy with the distribution of real images, thus leading to lower FID scores.

For a comprehensive understanding of the artifacts introduced by the refinement process, particularly from the DDIM inversion, we direct readers to Appendix B.3. Moreover, we provide a qualitative comparison between results with and without the refinement step in Appendix B.2, offering a more intuitive grasp of the refinement's impact on instance swapping.

### A.9 Analysis of DINO features

To better understand the signals provided by SD features, we introduce two variants to offset the limitations of DINO features. Both variants are designed to harness different aspects that contribute to improving the performance of DINO features. The first variant, dubbed Filter-ViT-B/14, incorporates a bilateral filter to enhance the smoothness of the features. The second variant, Ensemble-ViT-B/14, combines an early layer (layer 9) with the last layer to capture more spatial information.

As shown in Tab. 10, our handcrafted adjustments to DINO features result in an improved performance compared to the original DINOv2-ViT-B/14 model. This validates our strategies, underlining the importance of smoothness and spatial context in the dense correspondence task. However, it's worth noting that these adjustments offer only marginal improvements. This suggests that there are other valuable signals provided by SD features that we have yet to uncover, and it also reaffirms the efficacy of our fusion approach, Fuse-ViT-B/14. We will continue our exploration in future work.

Table 10: **Performance comparison of DINOv2 variants.** Each variant attempts to enhance the DINOv2 features by addressing its limitations. Filter-ViT-B/14 employs a bilateral filter for smoothness, while Ensemble-ViT-B/14 combines early and last layers to capture more spatial information. We also include the result of our method, Fused-ViT-B/14, for reference.

| Method | SPair-71k | | | PF-Pascal | | | TSS, PCK@0.05 | | | |
| | 0.01 | 0.05 | 0.10 | 0.05 | 0.10 | 0.15 | FG3DCar | JODS | Pascal | Avg. |
|---|---|---|---|---|---|---|---|---|---|---|
| DINOv2-ViT-B/14 | 5.8 | 40.0 | 55.4 | 61.1 | 77.3 | 83.3 | 82.8 | 73.9 | 53.9 | 72.0 |
| Filter-ViT-B/14 | 4.7 | 37.4 | 55.3 | 61.6 | 79.0 | 84.6 | 86.4 | 76.0 | 60.3 | 76.2 |
| Ensemble-ViT-B/14 | 6.4 | 40.7 | 56.1 | 62.6 | 79.7 | 86.1 | 84.7 | 74.2 | 59.7 | 74.8 |
| Fuse-ViT-B/14 | 8.2 | 47.5 | 62.9 | 72.1 | 86.0 | 90.6 | 94.3 | 73.2 | 60.9 | 79.7 |

### A.10 Quantitative analysis on features fusion

In this section, we aim to investigate the complementary nature of SD and DINO features through a quantitative analysis. By considering eight possible combinations of the performance of SD, DINO, and fused features, we explore how their fusion potentially enhances the overall effectiveness.

The hypothesis driving our investigation is that: 1) when SD features produce false matches, their feature distance to the correct matches could be marginally higher than that to the false matches; 2) Conversely, the DINO feature distance to the correct matches could be significantly smaller than that to the SD's false matches. Thus, by fusing these two features, the resulting representation could be pushed closer to the correct matches and further away from the false ones.

**Distribution of outcomes.** Tab. 11 shows the distribution of eight possible outcomes when SD, DINO, and fused features are used for generating correct correspondences. We notice that in scenarios where DINO or SD feature works individually and the other fails, but the fusion works (*i.e.*, notations 011 and 101) account for considerable proportions of cases. More importantly, these scenarios play the most crucial role in making the fused features work correctly. This finding supports our hypothesis about the complementary nature of SD and DINO features in terms of their fusion.

**Discussion on conditional probability.** Turning our attention to the conditional probabilities illustrated in Fig. 2, it is evident that the fusion approach enhances the overall probability of success. We can summarize this in four categories:

Table 11: **Distribution of eight possible outcomes for SD, DINO, and fused features** regarding their success (denoted by "✔") or failure (denoted by "✗") in generating correct correspondences, based on the PCK@0.10 measured on the SPair-71K test set. We underline the cases where fused is correct.

| SD | DINO | Fused | Notation | Interpretation | Ratio |
|----|------|-------|----------|----------------|-------|
| ✗ | ✗ | ✗ | 000 | SD, DINO, and Fused all fail | 26.30% |
| ✗ | ✗ | ✔ | 001 | SD and DINO fail, Fused correct | 2.90% |
| ✗ | ✔ | ✗ | 010 | SD and Fused fail, DINO correct | 5.82% |
| ✗ | ✔ | ✔ | 011 | SD fails, DINO and Fused correct | 9.94% |
| ✔ | ✗ | ✗ | 100 | DINO and Fused fail, SD correct | 4.57% |
| ✔ | ✗ | ✔ | 101 | DINO fails, SD and Fused correct | 10.76% |
| ✔ | ✔ | ✗ | 110 | Fused fails, SD and DINO correct | 0.38% |
| ✔ | ✔ | ✔ | 111 | SD, DINO, and Fused all correct | 39.28% |

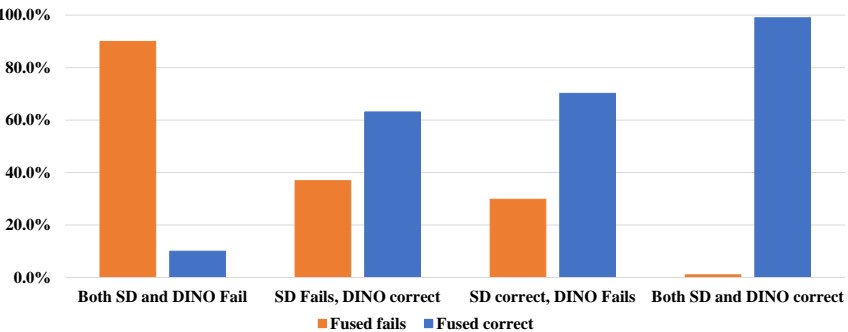

Figure 2: **Conditional probabilities illustrating the performance of fused features** under different scenarios with SD and DINO features.

1. When both SD and DINO features fail to produce correct correspondences (the 1st group), the fused feature manages to rectify this $10.0\%$ of the time.
2. When SD features fail but DINO features succeed (the 2nd group), the fused feature successfully produces correct correspondences $63.1\%$ of the time.
3. When SD features succeed but DINO features fail (the 3rd group), the fused feature achieves a success rate of $70.2\%$.
4. When both SD and DINO features succeed (the 4th group), the fused feature fails very rarely, with a failure rate of only $1.0\%$.

These results confirm the complementary nature of the SD and DINO features. Even when both SD and DINO individually fail to generate correct correspondences, the fused feature can occasionally correct these errors. On the other hand, if either SD or DINO feature is correct, the fused feature is highly likely to also be correct. Finally, when both SD and DINO are correct, the fused feature almost always retains this accuracy. These observations lend further credibility to the efficacy of our feature fusion approach in augmenting the success rate of correct correspondence generation.

**Analysis of relative distance.** To delve deeper into why feature fusion succeeds in scenarios 011 and 101, we introduce a metric called "relative distance". Given a specific point in the source image and a corresponding point in the target image, we calculate the distances from the source point to all points in the target image. We then normalize the distance to the target point, using the minimum and maximum distances. This normalized distance, or "relative distance", provides a measure of feature matching confidence, with smaller distances indicating higher confidence.

Tab. 12 illustrates the relative distances for SD and DINO features under scenarios 011 and 101. In the case of scenario 011, the relative distance of the correct match under SD features is 0.156, which is lower than the average relative distance of 0.218 for correct matches. This underscores that SD features exhibit a higher confidence for correct matches under this scenario, which lends credence to our hypothesis that when SD features produce false matches, the feature distance to the correct matches tends to be marginally lower.

Table 12: **Relative distances for SD and DINO features** for both correct matches and false matches under scenarios 011 and 101, compared with the average situation. The false matches indicate the matches falsely identified by the other feature.

| | SD Features | | DINO Features | |
|---|---|---|---|---|
| Scenario | Correct Match | False Match | Correct Match | False Match |
| 011 | 0.156 | – | – | 0.328 |
| 101 | – | 0.248 | 0.141 | – |
| Average | 0.218 | 0.200 | 0.198 | 0.239 |

Furthermore, for the DINO features in the same scenario, the relative distance of SD's false match is 0.328, significantly higher than the average situation (0.239). This observation implies that DINO features generally exhibit lower confidence in matches that SD falsely identifies, thereby supporting our hypothesis that the DINO feature distance to the correct matches could be significantly smaller than that to the false matches produced by SD.

In scenario 101, the roles of SD and DINO features are reversed, but the trends observed in scenario 011 continue to hold true. The relative distance of the false match under SD features is 0.248, while the relative distance of the correct match under DINO features is 0.141. Once again, these results strengthen our belief in the complementary nature of SD and DINO features, where the strengths of one feature compensate for the weaknesses of the other.

These findings, therefore, shed light on why fusion works effectively under scenarios 011 and 101 and provides empirical evidence for our hypothesis of the complementary nature of SD and DINO features.

# B   Additional results

We provide additional qualitative and quantitative results in this section.

## B.1   Dense correspondence

We present additional qualitative result for both SPair-71k ( Figs. 3 and 4) and DAVIS ( Fig. 5) datasets.

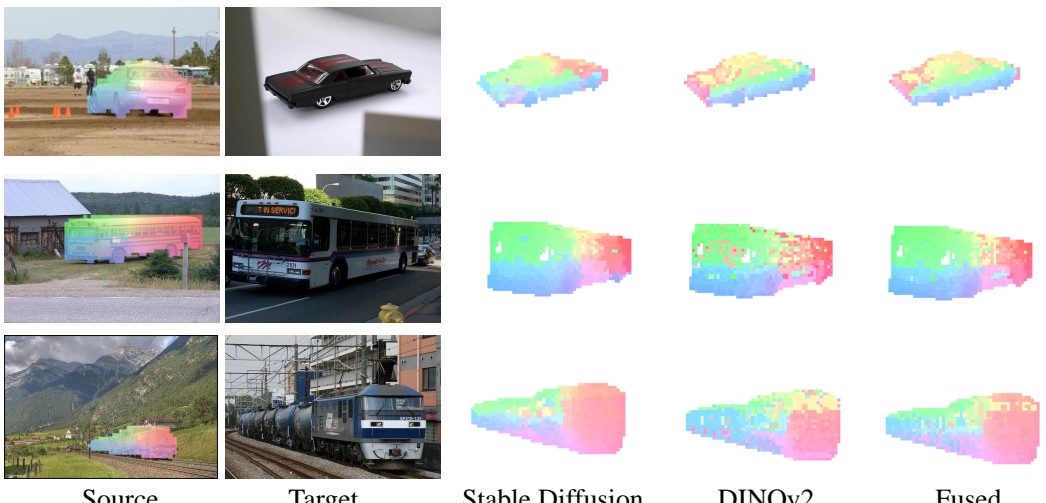

| Source | Target | Stable Diffusion | DINOv2 | Fused |

Figure 3: **Dense correspondence on SPair-71k** (rigid).

## B.2   Instance swapping

We present qualitative comparison of instance swapping on Paint-by-Exaple [13] dataset in Fig. 6.

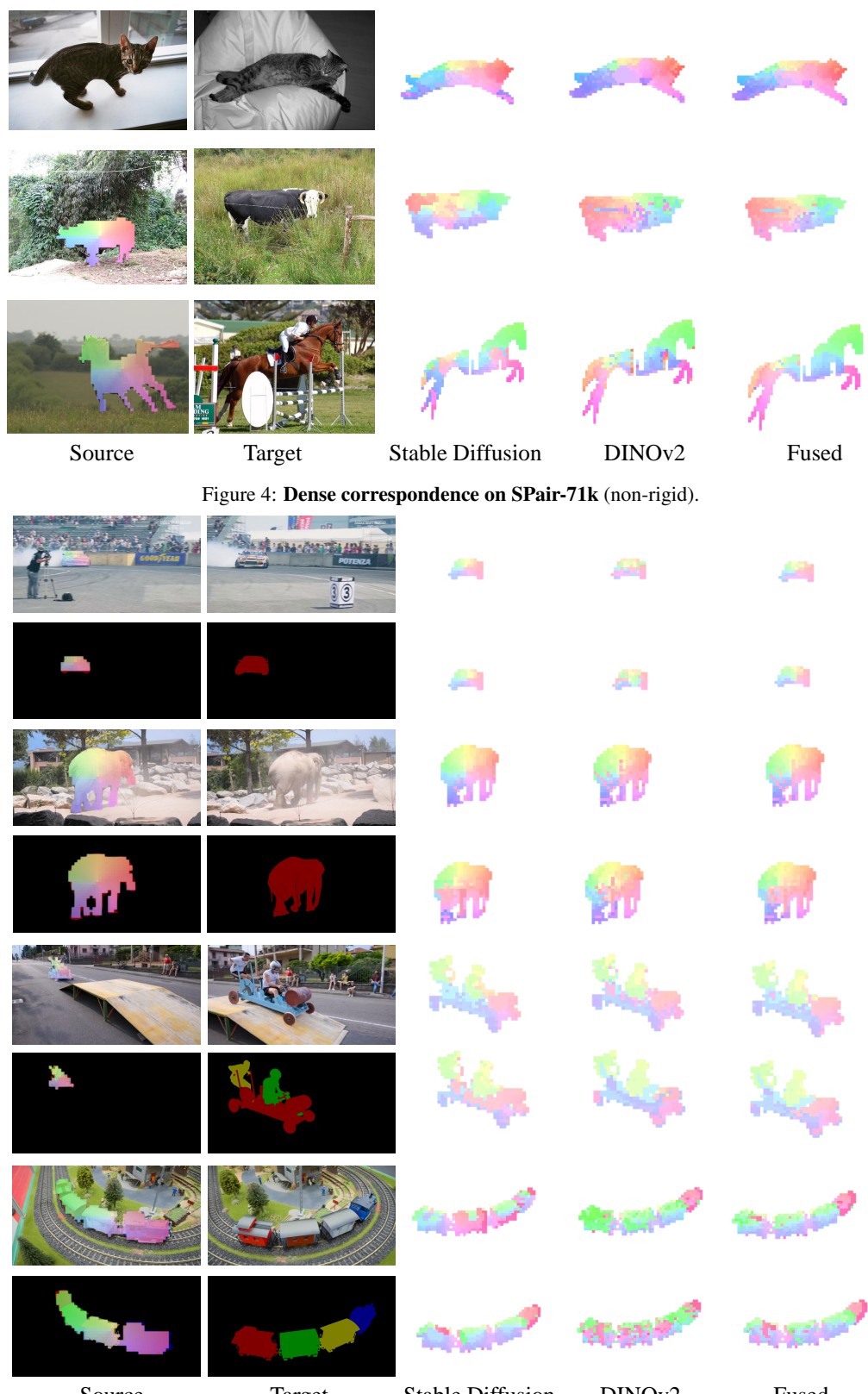

Source       Target       Stable Diffusion       DINOv2       Fused

Figure 4: **Dense correspondence on SPair-71k** (non-rigid).

Source       Target       Stable Diffusion       DINOv2       Fused

Figure 5: **Dense correspondence on DAVIS.** The results are for both real images and annotation maps as inputs.

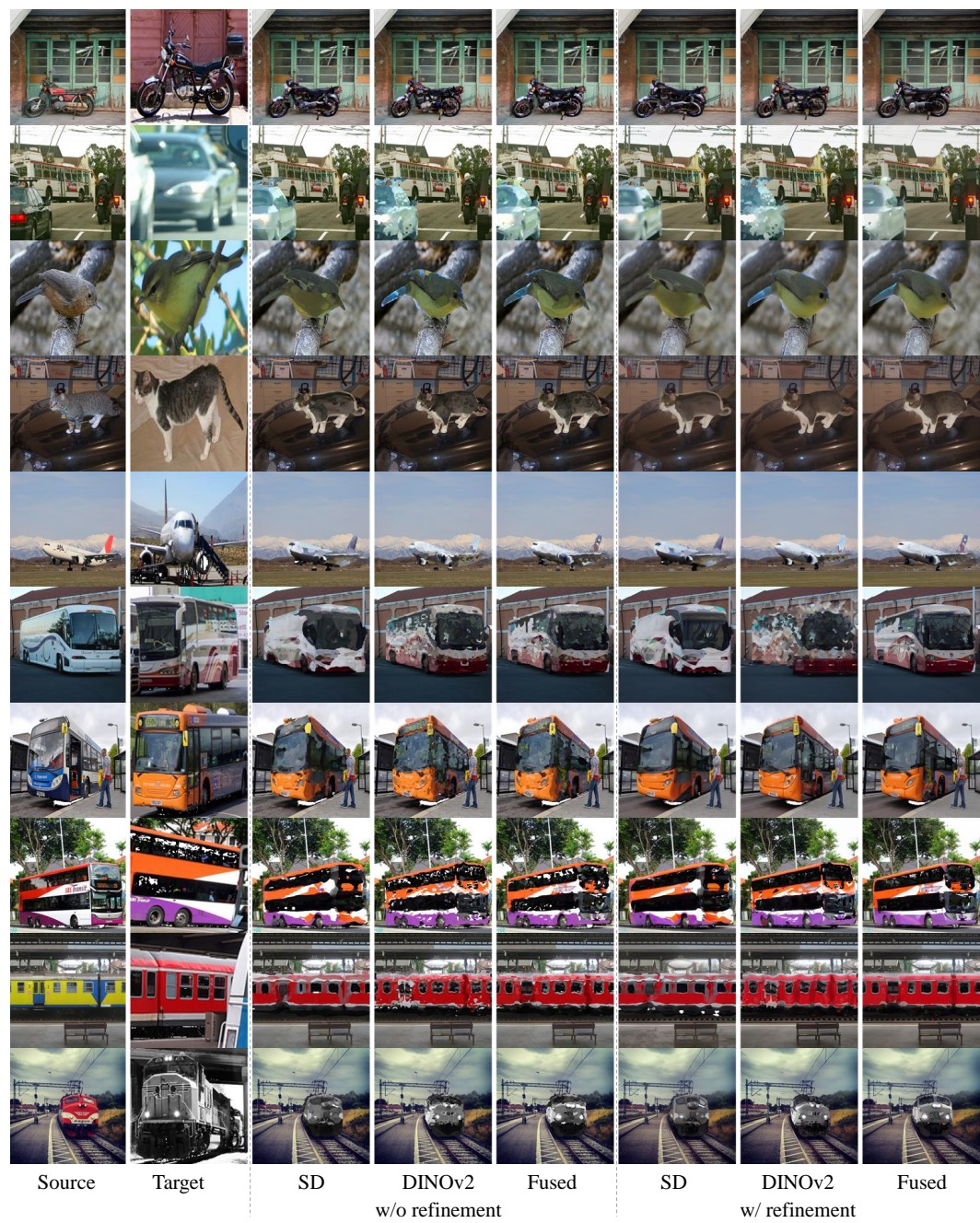

| Source | Target | SD | DINOv2 | Fused | SD | DINOv2 | Fused |
|--------|--------|----|--------|-------|-----|--------|-------|
|        |        |    | w/o refinement | |     | w/ refinement | |

Figure 6: **Qualitative comparison on instance swapping.** We present the results of both w/o and w/ refinement.

## B.3 Failure cases

We present two types of failure cases in Fig. 7.

## B.4 Results for CUB-200 dataset

We evaluated our method on the CUB-200 [11] dataset, which comprises over 200 fine-grained bird categories. With the ASIC protocol [5], we compare our approach with various methods on the first three categories subsets of the dataset, in Tab. 13.

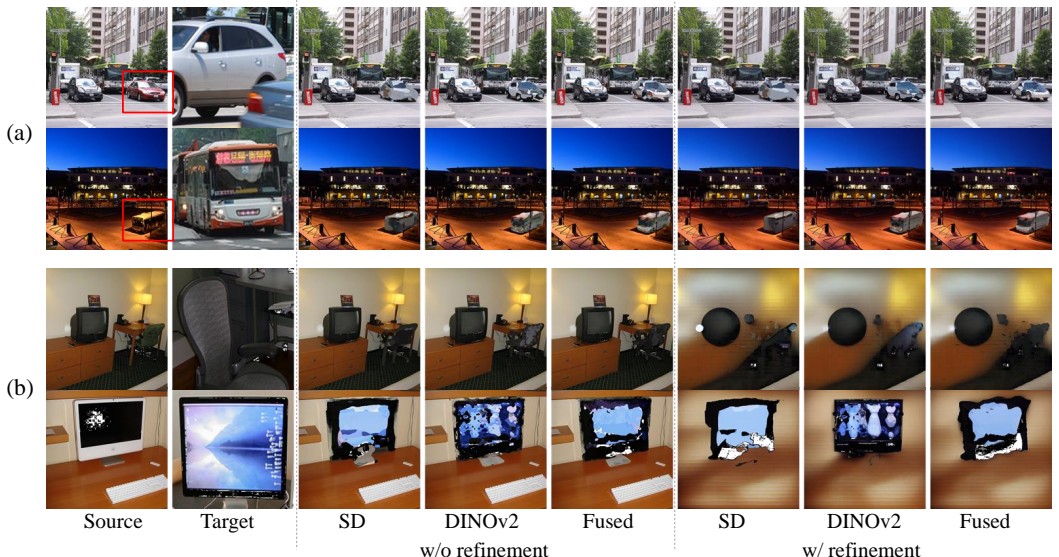

| | | | | | | |
|---|---|---|---|---|---|---|
| Source | Target | SD | DINOv2 | Fused | SD | DINOv2 | Fused |
| | | | w/o refinement | | | w/ refinement | |

Figure 7: **Failure cases.** (a) When the relative size of instance of interest is tiny in the image, (b) Artifacts introduced by DDIM inversion.

Table 13: **Quantitative comparison on the CUB-200 subsets.** $U^T$ denotes the task-specific unsupervised methods, and $U^N$ denotes the nearest-neighbor-based unsupervised methods. The best performance in **bold**.

| | Method | PCK@0.1 | PCK@0.05 | PCK@0.01 |
|---|---|---|---|---|
| $U^T$ | VGG+MLS [1] | 25.8 | 18.3 | - |
| | DINOv1+MLS [1, 3] | 67.0 | 52.0 | - |
| | ASIC [5] | 75.9 | 57.9 | - |
| $U^N$ | DINOv1+NN [2] | 68.3 | 52.8 | - |
| | DINOv2-ViT-B/14 | 80.0 | 65.6 | 12.7 |
| | Stable Diffusion **(Ours)** | 61.4 | 46.7 | 8.7 |
| | Fused ($\alpha = 0.5$) **(Ours)** | 79.2 | 64.6 | **14.3** |
| | Fused ($\alpha = 0.8$) **(Ours)** | **80.3** | **66.2** | 12.2 |

Our approach outperforms other methods across all thresholds, especially showing substantial improvement on the challenging PCK@0.01 metric. Our fusion approach again yields superior results compared to each feature, DINOv2 and Stable Diffusion. validating the efficacy of our approach.

## C  Data License

In our paper, all utilized images are either sourced from the following publicly available datasets or generated for research purposes. Here we provide the data sources and their corresponding licenses.

1. **DAVIS:** The Densely Annotated VIdeo Segmentation (DAVIS) dataset is extensively used in our work. The dataset can be accessed at https://davischallenge.org/. It is distributed under the BSD License (https://github.com/fperazzi/davis-2017/blob/main/LICENSE).

2. **PASCAL-VOC:** The PASCAL Visual Object Classes (VOC) dataset is another source of our images. The dataset can be accessed at http://host.robots.ox.ac.uk/pascal/VOC. The PASCAL VOC data is provided under the Flickr terms of use https://www.flickr.com/help/terms.

3. **MSCOCO:** Certain images used in our work are taken from the Microsoft Common Objects in Context (MSCOCO) dataset. This dataset can be accessed at https://cocodataset.org/. The license for MSCOCO can be found at https://creativecommons.org/licenses/by/4.0/.

For all generated images in this paper, we affirm that they are exclusively produced for the purpose of academic research.

## D    Broader Impact

Our work has presented an innovative representation for semantic correspondence, achieved by examining the properties of SD features and DINOv2 features, and applying a fusion method to leverage their complementary strengths. Although the emphasis of this paper was not on the adaptation of task-specific methods, our approach can potentially serve as a valuable addition to such techniques.

For instance, our method could be integrated as a feature extractor within existing systems, such as those proposed in [5, 8]. We suggest that our method, if used in place of their existing feature extractors, might offer new opportunities for enhancement. It may open avenues for potential improvements to existing task-specific approaches.

However, like any technology, misuse could lead to negative implications. High-quality object swapping might be used maliciously to create deceptive or misleading imagery. As we continue our research, we are committed to considering both the positive and potential negative impacts of our work, striving to contribute responsibly to the advancement of computer vision technology.

## E    Discussions

We acknowledge that the models utilized in our study, namely Stable Diffusion and DINOv2, may have been exposed to some of the images from PF-PASCAL, SPair-71k, CUB-200, etc., during their training phase. However, considering the colossal scale of the datasets used for training these models – 2B images for Stable Diffusion and 142M for DINOv2 – the chance that the models could distinctly remember a small number of images ($\sim$1K) from these datasets is extraordinarily low, less than $0.00005\%$ and $0.0007\%$ for SD and DINOv2, respectively. Moreover, the vast difference in training objectives between these models and our tasks further minimizes any potential influence. Thus, any potential impact on our study's results from this overlap is likely negligible.