# OpenReview forum: "A Tale of Two Features: Stable Diffusion Complements DINO for Zero-Shot Semantic Correspondence"
_NeurIPS.cc/2023/Conference — NeurIPS 2023 poster_

### Official Review · Reviewer_YopU · 2023-07-06

**Soundness:** 3 good
**Presentation:** 4 excellent
**Contribution:** 3 good
**Rating:** 7
**Confidence:** 4

**Summary:**

This paper exploits Stable Diffusion (SD) features for semantic and dense correspondence tasks. The authors first conduct evaluations of SD features and found that SD features provide high-quality spatial information but sometimes inaccurate semantic matches. This paper further shows that such SD features are complementary to DINOv2 features which provide sparse but accurate matches. By fusing SD and DINOv2 features with a simple weighted concatenation strategy, this paper achieves significant performance gains over state-of-the-art methods on benchmark datasets, e.g., SPair-71k, PF-Pascal, and TSS.

**Strengths:**

- **Extensive evaluation and analysis.** This paper conducts lots of experiments and visualizations to analyze the behavior of Stable Diffusion and DINOv2 features. The experiments are also well-organized, with a smooth logic flow, which makes it easy for readers to follow.
- **Strong zero-shot performance.** By simply combining the Stable Diffusion and DINOv2 features, this paper shows strong zero-shot performance of the semantic correspondence task, outperforming previous task-specific methods by a large margin.
- **The message shown in this paper could be interesting to the community.** Unlike previous methods that mostly focus on studying single image diffusion models, this paper studies the properties of diffusion model features across different images in the context of semantic correspondence. This paper shows that the publicly available Stable Diffusion and DINOv2 models contain rich information to solve the semantic correspondence task, even outperforming task-specific models in the zero-shot setting. This would be a strong message and might stimulate future work along this direction.



**Weaknesses:**

It's unclear how the proposed method handles outliers in the correspondences. Since the correspondences are obtained with nearest neighbor search, which might be less robust for occlusion or out-of-frame points.



**Questions:**

- I am wondering what might lead to the different behaviors of Stable Diffusion features and DINOv2 features? A more in-depth discussion might make this paper stronger. Could the authors further comment on this?
- As also discussed in the limitation section, the spatial resolutions of Stable Diffusion features and DINOv2 features are both relative low, which might hurt the performance for fine-grained details. Besides Stable Diffusion, there are also other diffusion models that directly work on the original pixel space instead of the downsampled latent space, would these kind of diffusion models potentially alleviate this issue? I am not asking for such experiments, but just wondering the authors' opinion on this.

**Limitations:**

Yes, the limitations are discussed in the paper.

---

> ### Author Rebuttal · Authors · 2023-08-10
>
>
> **W1:** Handle outliers in the correspondences.
>
> **A:** Our method primarily focuses on analyzing properties of each feature and their fusion, and thus we adopt a very simple matching method (i.e., NN search) without using any matching priors or templates. Despite this, our method achieves outperforming/competitive accuracy on benchmark datasets (Table 3 and 4 in the main paper). However, we believe certain methods that are designed specifically for outliers in the correspondence task (OmniMotion [1] for occlusion, and PWarpC [2] for unmatchable points) can benefit from our proposed fused features. We consider integrating these features into those systems or a novel fusion architecture which identifies occluded points or outliers to be an important future direction for our work.
>
> ------
>
> **Q1:** Causes of the complementary properties of SD and DINOv2.
>
> **A:** Please refer to the "General Response Q&A section".
>
> ------
>
> **Q2:** Potential of pixel-space diffusion models for overcoming limitations in feature resolution.
>
> **A:** While we agree that this seems like a promising direction, there may be a few challenges. Pixel-space diffusion models typically operate through a coarse-to-fine methodology. They start with an initial low-resolution, often 64x64, text-to-image diffusion process, followed by (several) subsequent upsample diffusion process(es). Extracting a feature map larger than 64x64 would necessitate relying on the upsample diffusion model. However, while the upsample diffusion models are crucial for image clarity, they do not necessarily need to comprehend the inherent structure of the image. As such, these models might contain less semantic information than the text-to-image diffusion models, potentially limiting its efficacy in semantic correspondence tasks.
>
> Additionally, from our investigation of competitive diffusion models in the current landscape, it appears that none of them simultaneously meet the conditions of 1) being open-sourced, 2) operating within the pixel-space, and 3) offering feature maps with a resolution higher than 64x64. This makes it challenging for us to test the above idea, though it certainly is interesting.
>
> |Model|FID on I-Net↓|Open-sourced|Pixel-space|Large f_map|
> |-|:-:|:-:|:-:|:-:|
> |GLIDE|12.24|||❌|
> |DALLE2|10.39|❌|||
> |SD|8.32||❌||
> |Imagen|7.27|❌|||
> |eDiff-I|6.95|❌|||
> |ERNIE-ViLG2.0|6.75|❌|❌||
> |RAPHAEL|6.61|❌|❌||
>
> Table 1. Survey of competitive diffusion models.
>
> As a potential fix for the spatial resolution limitation, one could train an additional projection layer. This layer, when trained on top of both the input image and its corresponding low-resolution features, might offer a bridge between spatial granularity and semantic depth. We consider this a promising direction for future exploration.
>
> -----
> **References**
>
> [1] Tracking Everything Everywhere All at Once, Wang et al. 2023
>
> [2] Probabilistic Warp Consistency for Weakly-Supervised Semantic Correspondences, Truong et al. 2022

---

> > ### Comment · Reviewer_YopU · 2023-08-13
> >
> > Thanks the authors for the detailed response, and I am happy to increase my rating to Accept.

---

> > > ### Author Response · Authors · 2023-08-13
> > >
> > > Thank you for your feedback. We appreciate your comments and will further refine our paper.

---

### Official Review · Reviewer_NMVR · 2023-07-06

**Soundness:** 3 good
**Presentation:** 3 good
**Contribution:** 3 good
**Rating:** 7
**Confidence:** 5

**Summary:**

This paper, for the first time, proposes a novel method to fuse Stable Diffusion features and DINOv2 features to obtain robust feature representation that readily surpasses the SOTA semantic correspondence work without further training, rather simply adopting Winner Takes All yields SOTA correspondence performance. Interestingly, with such correspondences, high-quality object swapping is also enabled without task-specific fine-tuning.

**Strengths:**

1. Novel idea, approach and thorough analysis.

2. Motivation is really good and the paper was easy to read and understand.

3. Impressive performance.

4. Good application (swapping)

**Weaknesses:**

1. This is more like a question rather than a weakness. In figure 1, the visualization shows that a cat given as a source image, the proposed approach finds dense "correspondence" to dog, horse and motorcycle. Is this really a correspondence? To me, it feels like this is more like  foreground segmentation.  From the colors of the features, iI could tell that the facial part of cat is matched to the frontal part of motorcycle and facial part of other animals. This is very interesting. Why do you think this visualization is obtained even though strictly speaking, there should not be correspondences?

2. Why is it that on SPair-71k, the proposed method yields leading results, but not for  PF-PASCAL and TSS? As the authors state, PF-PASCAL and TSS are less challenging datasets, but results seem to show the opposite.

3. Missing baselines : for supervised ones, CATs++ (extension of CATs) and IFCAT from the same author achieve better performance. Also, PWarpC is a work that adopts unsupervised learning for semantic correspondence. I would suggest including this as well. It is better to compare with them as well. I don't think it would be a weakness even if the proposed method does not attain SOTA, since the contribution lies more on the fusion of features and their analysis.

4. In implementation detail, it says the input resolution is 960 x 960. What was the evaluation keypoint annotation resolution? The input resolution, evaluation resolution and many other factors related to resolutions have high influence on performance in this task. This is addressed by PwarpC (CVPR'22) and CATs++ (TPAMI). So if the evaluation is performed at higher resolution than other works, the comparison may not be fair at all. This needs to be clarified.

**Questions:**

See the weaknesses above. If they are adequately addressed in the rebuttal, I think this paper will be a very strong paper. My rating is solely based on the performance part, which  if clarified, then I am happy to increase my rating to accept this paper.

**Limitations:**

Limitations are adequately discussed.

---

> ### Author Rebuttal · Authors · 2023-08-10
>
> **W1**: Clarity on the visualization of Fig. 1.
>
> **A**: We provide the challenging cross-category semantic correspondence that matches semantically related or geometrically similar parts across different object categories or even domains. The Neural Best-Buddies [1] and DINOv2 paper also visualizes those examples.
>
> As in Fig. 1, our method matches objects even at different poses, suggesting that diffusion features contain information related to semantic/pose/shape. This is more information than needed for foreground segmentation. We hypothesize that diffusion features contain high-level semantic and structural information which can relate conceptually different objects eg, cats and motorcycles.
>
> ---
> **W2**: Clarity on the performance of PF-Pascal and TSS datasets.
>
> **A:** Our method achieves leading results on PF-Pascal and TSS datasets, specifically:
>
> - **The best results on PF-Pascal dataset among the *unsupervised* methods**, with 49.17%, 18.62%, and 6.46% relative improvement on PCK@0.05, 0.10, and 0.15, respectively (see Table 1 below). Additionally, if we use a simple form of supervision for our fused features (by learning a single projection layer with the PF-Pascal training set), **our method performs similarly to other supervised methods on PF-Pascal**. If we were to fine-tune the backbone networks (SD or DINOv2) like other supervised approaches, we would expect to see even bigger improvements.
> - **The best results on TSS among *unsupervised nearest neighbors* methods** and is near SOTA for *unsupervised* methods including those with task specific networks and losses (see Table 2 below). While Semantic-GLU-Net performs better on TSS, it is important to note that Semantic-GLU-Net significantly underperforms our fused features on SPair-71k (23.5 vs our 62.9 for PCK@0.1) and PF-Pascal (72.5 vs our 86.0 for PCK@0.1).
>
> |Technique|Method|PCK@0.15|PCK@0.10|PCK@0.05|
> |-|:-|:-:|:-:|:-:|
> |Unsupervised|CNN-Geo [43]|41.0|69.5|80.4|
> ||Glu-Net [57]|42.2|69.1|83.1|
> ||Semantic-Glu-Net [60]|48.3|72.5|85.1|
> ||Stable Diffusion (**Ours**)|*63.2*|*77.7*|*86.3*|
> ||Fuse-ViT-B/14 (**Ours**)|**72.1**|**86.0**|**90.6**|
> ||
> |Supervised|PWarpC-NC-Net|79.2|92.1|95.6|
> ||CATs++* |*84.9*|*93.8*|96.8|
> ||IFCAT*|**88.0**|**94.8**|**97.9**|
> ||Fuse-ViT-B/14 trained projection layer (**Ours**)|80.9|93.6|*96.9*|
>
> Table 1. PF-Pascal Performance. * denotes finetuned backbones. In a category, **bold** is best, *italics* is second best.
>
> |Technique|Method|FG3DCar|JODS|Pascal|Avg.|
> |-|-|:-:|:-:|:-:|:-:|
> |Unsupervised TS|CNN-Geo [43]|90.1|*76.4*|56.3|74.4|
> ||PARN [23]|89.5|75.9|*71.2*|78.8|
> ||GLU-Net [57]|93.2|73.3|71.1|79.2|
> ||Semantic-GLU-Net [60]|**95.3**|**82.2**|**78.2**|**85.2**|
> ||
> |Unsupervised NN|DINOv2|82.8|73.9|53.9|72.0|
> ||Stable Diffusion (**Ours**)|93.9|69.4|57.7|77.7|
> ||Fuse-ViT-B/14 (**Ours**)|*94.3*|73.2|60.9|*79.7*|
>
> Table 2. TSS Performance.  TS is task-specific, NN is nearest-neighbors. **Bold** is best, *italics* is second best. Fuse-ViT-B/14 (**Ours**) performs best among Unsupervised NN methods and second best among all Unsupervised methods (see note above regarding Semantic-GLU-Net).
>
> **With regards to the dataset difficulty**, weaker methods can often perform better on easier datasets by exploiting specific biases or limitations. Specifically,
> - **TSS** contains a limited number of object classes mostly under rigid transforms. Having a template-based parametric approach [2] can achieve good performance on TSS.
> - **PF-Pascal** contains image pairs with the same viewpoint, and does not require an intrinsic understanding of the instance's structure and orientation.
> - **SPair-71k**, in contrast, contains image pairs with more diverse viewpoint, scale, occlusion, and truncation than PF-PASCAL and TSS, is known to be more challenging and thus more informative as an evaluation.
>
> That said, **achieving strong performance on SPair-71k is particularly commendable**. Our zero-shot setting already achieves comparable performance with the supervised SOTA on SPair-71k. With a simple form of supervision to our fused features, our method surpasses supervised SOTA (ours 78.2 vs. IFCAT* 64.4).
>
> ---
> **W3**: Additional baselines.
>
> **A:** We include PWarpC-CATs in Table 4 in the main paper but didn’t include its weakly-supervised variations due to space limit. Please see Tab. 1 in R2-fsqe's W1 response, for a comparison with CATs++, IFCAT, and PWarpC; this show that we still attain SOTA in the challenging SPair-71k datasets under supervised setting. We will include these baselines in the revised version.
>
> ---
> **W4**: Effect of the keypoint annotation resolution and the input image resolution.
>
> **A:** Our evaluation keypoint annotation resolution is 840, employed for all NN-based methods. Table 3 provides the PCK results for SPair-71k 20-samples subsets at different input resolutions and annotation resolutions (the min input resolution is restricted by the SD model). Similar to PWarpC's Table 1, using different annotation resolutions only marginally affects the accuracy. On the other hand, as in CATs++, we also observe that input image's resolution matters more, especially under stricter constraints (PCK@0.05 and 0.01).
>
> |Input Image Reso.|Annotation Reso.|0.10|0.05|0.01|
> |-|-|:-:|:-:|:-:|
> |960|840|63.28|47.61|8.32|
> |(feat map 60*60)|ori.|62.80|47.36|8.11|
> ||256|62.54|47.58|8.36|
> ||
> |512|840|61.66|40.73|4.58|
> |(feat map 32*32)|ori.|61.58|40.43|4.28|
> ||256|61.51|40.35|4.40|
>
> Table 3. PCK performance of fuse features under different input image resolution and keypoint annotation resolution on SPair-71k.
>
> Comparing with CATs++ under the same settings (512 input and 256 annotation resolution), our method still performs better, notably in PCK@0.10, with ours 61.51 vs theirs 59.9 (number taken from their Fig. 12a).
>
> ---
> **References**
>
> [1] Neural Best-Buddies: Sparse Cross-Domain Correspondence, Aberman et al. 2018
>
> [2] GLU-Net: Global-Local Universal Network for Dense Flow and Correspondences, Truong et al. 2020

---

> > ### Comment · Reviewer_NMVR · 2023-08-11
> >
> > Thanks for the thorough responses.
> > As I don't find any other major concerns, I will increase the rating.

---

> > > ### Author Response · Authors · 2023-08-11
> > >
> > > Thank you! We appreciate your comments and will revise this paper accordingly.

---

### Official Review · Reviewer_fh35 · 2023-07-06

**Soundness:** 1 poor
**Presentation:** 2 fair
**Contribution:** 2 fair
**Rating:** 6
**Confidence:** 3

**Summary:**

The paper explores the use of Stable Diffusion (SD) features for dense correspondence. The authors investigate the potential of SD and DINOv2 features and show some complementarity. SD features provide high-quality spatial information but sometimes inaccurate semantic matches while DINOv2 features offer sparse but accurate matches. The authors show that averaging the features from the two models might achieve strong performances for dense correspondence. The fused features are evaluated using a zero-shot approach, where no specialized training is performed for the correspondence task, and nearest neighbors are used for evaluation. Surprisingly, the fused features outperform state-of-the-art methods on benchmark datasets like SPair-71k, PF-Pascal, and TSS. The authors also demonstrate that these correspondences enable interesting applications such as instance swapping between images while preserving object identities.

**Strengths:**

**In-depth Qualitative Analysis** The paper conducts a detailed qualitative analysis of the Stable Diffusion (SD) features and DINOv2 features, shedding light on their respective strengths and weaknesses. This analysis provides valuable insights into the semantic and texture relevance of these features, highlighting their distinct properties.

**Extensive Experiments** The paper presents extensive experimental results on benchmark datasets, including SPair-71k, PF-Pascal, and TSS. The authors report significant gains compared to SOTA methods.

**Application of Instance Swapping** The paper well illustrate the complementarity of the feature on this task.

**Weaknesses:**

**Lack of Strong Quantitative Assessments** While the paper provides an in-depth qualitative analysis of the complementarity between SD and DINOv2 features, it lacks strong quantitative assessments to support these claims. The true added value of the paper should have been a robust quantitative evaluation showcasing in particular the non-redundancy of the SD and DINOv2 features. This absence limits the overall impact and credibility of the proposed approach.

**Incomplete Figure 2 ?** There appears to be an issue with Figure 2, as the bottom illustration is missing despite being mentioned in the legend.

**Lack of Clarity in PCA Aggregation** The authors mention the use of PCA for aggregating SD features, but the explanation provided is not clear enough. The authors mention computing PCA across the pair of images for each layer of the feature map, followed by upsampling features from different layers to the same resolution to form the ensembled feature. I however have trouble understanding clearly what is the workflow here.

**Questions:**

To provide a fair evaluation, it would be interesting to have the performances of other fused features based on DINO varaitions such as DINOv1 + SD or EsViT[1] + SD.

[1] Chunyuan Li et al. “Efficient Self-supervised Vision Transformers for Representation Learning”, ICLR22

**Limitations:**

The authors underline the technical limitation of their work.

---

> ### Author Rebuttal · Authors · 2023-08-10
>
>
> **W1**: Quantitative assessments on the non-redundancy of SD and DINOv2 features.
>
> **A**: We include several quantitative evaluations that underscore the non-redundancy and distinctiveness of SD and DINOv2 features. Specifically, we provide: *1) quantitative error analysis on fused and individual features; 2) evidence of non-redundancy across datasets; 3) smoothness analysis on the TSS flow fields; and 4) the correspondence performance of the fused and independent features.* Details from these evaluations are provided as follows:
>
> 1. **Quantitative error analysis on fused and individual features:** In **Supplemental A.7**, we provide an in-depth quantitative analysis on feature fusion. It includes:
>     - Distribution of outcomes, considering the performance of SD, DINOv2, and fused features -- showing the considerable proportion of cases where only one features succeeds;
>     - A detailed discussion on the conditional probability of the performance of fused features under different scenarios of SD and DINOv2 -- showing fusion can helps rectify when one or both features fail;
>     - A relative distance analysis in certain senarios -- further sheding light on how the fused features helps under certain circumstances.
>
> 2. **Non-redundancy across datasets and PCK levels:** We present additional evidence of the non-redundancy of SD and DINOv2 features in Table 1, which details the error distribution for these two features on SPair-71k and PF-Pascal benchmarks at three different PCK levels. As shown in this table, in 20~30% of total cases under most settings, one feature succeeds where the other fails; this suggests that they have a substantial amount of non-redundant information.
>
> |||SD, DINO fails|SD fails, DINO correct|SD correct, DINO fails|SD, DINO correct|
> |-|-|:-:|:-:|:-:|:-:|
> |SPair-71k|PCK@0.15|21.68|15.70|13.95|48.67|
> ||PCK@0.10|29.20|15.76|15.33|39.66|
> ||PCK@0.05|44.50|14.20|15.81|25.50|
> ||
> |PF-Pascal|PCK@0.15|5.60|8.27|11.12|75.01|
> ||PCK@0.10|9.99|9.68|12.72|67.62|
> ||PCK@0.05|27.07|11.98|16.78|44.17|
>
> Table 1. Distribution of outcomes under different datasets and PCK levels.
>
> 3. **Smoothness analysis on the TSS flow fields:** Table 2 in the main paper offers a quantitative analysis of smoothness on computed flow fields and suggests that SD features produce dense correspondences which are much smoother than those from DINOv2 features; this strongly suggests that SD features contain more spatial information than DINO features.
> 4. **Correspondence performance of fused and individual features:** In Table 3 of the main paper, the improved quantitative performance of our fusion results shows that these features have non-redundant elements (otherwise fusing them would not provide a quantitative improvement). Our fusion result leads to a 13% improvement on SPair over either individual feature. Additionally, Table 3 reports the performance of different features on per-category subsets, with accompanying analysis in Section 4.1. In particular, several categories demonstrate that DINO and SD have drastically different performance. Aero, Bike, Boat, Car, Dog, Horse, Motor, Person, Sheep all perform better with DINO features, while Cow, Plant, Train, TV perform better with SD features. Specifically, the categories that SD performs better on tend to have less texture signal (e.g. TV, Plant).
>
> We also provide qualitative analysis to support our claim in: 1) Section 3.3 which showcases extensive qualitative experiments highlighting multiple distinctions between the two features; 2) Supplemental B.1 where we provide additional visual results to showcase the complementary.
>
> We will highlight the quantitative analysis more clearly in the updated main paper.
>
> ------
> **W2**: Clarity on the grouping of Figure 2.
>
> **A:** We divide Figure 2 into two parts with a dotted line. The top two rows show the visualization of PCA-computed features and the bottom two rows show K-Mean clustered features. We will make the figure more clear in an updated version.
>
> ------
> **W3**: Clarity on the details of PCA aggregation.
>
> **A:** Here is a more detailed explanation on how to aggregate SD features using PCA.
>
> We first extract the $i^{th}$ layer’s features for source and target images, $\{f^s\_i\}\_{i\in[2,5,8]}$ and $\{f^t\_i\}\_{i\in[2,5,8]}$. Next, we concatenate each layer’s source feature and target feature and compute PCA together: $\{\tilde f^s\_i, \tilde f^t\_i = PCA(f^s\_i || f^t\_i)\}\_{i\in[2,5,8]}$. Then we gather each layer’s dimension-reduced features $\{\tilde f^s\_i\}\_{i\in[2,5,8]}$ and $\{\tilde f^t\_i\}\_{i\in[2,5,8]}$, and upsample them to the same resolution to form the final SD feature $\tilde f^s$ and $\tilde f^t$.
>
> We will include these details in the main paper.
>
> ------
> **Q1**: Performance of other DINO variations.
>
> **A:** The pre-trained EsViT model checkpoint is currently inaccessible due to public access restrictions (issue #27 in the EsViT GitHub repo). Thus, we tried other DINO variations - iBOT [1], and DINOv1. As shown in Table 2, both DINOv1 and iBOT perform significantly worse than DINOv2. Surprisingly, a zero-shot fusion with SD slightly decreases the overall performance. We hypothesize that this may be due to the relatively weak performance of DINOv1 and iBOT; if these features are strictly worse than SD features, they may only contribute noise to the zero-shot fusion results. A learned projection would enable fusion to ignore features that decrease the overall performance and at least perform similarly to SD by itself.
>
> |Method|PCK@0.10|PCK@0.05|PCK@0.01|
> |-|:-:|:-:|:-:|
> |DINOv1-vitb16|33.17|19.93|2.43|
> |iBOT-vitb16|38.85|23.90|2.63|
> |DINOv2-vitb14|55.15|39.66|6.12|
> ||
> |Stable Diffusion|56.18|42.80|6.79|
> |Fuse-DINOv1-vitb16|51.79|37.50|5.34|
> |Fuse-iBOT-vitb16|55.11|38.99|4.85|
> |Fuse-DINOv2-vitb14|63.28|47.61|8.32|
>
> Table 2. Comparison with DINOv2 and two other variants on SPair-71k.
>
> ------
> **References**
>
> [1] iBOT: Image BERT Pre-Training with Online Tokenizer, Zhou et al. 2022

---

> > ### Author Response · Authors · 2023-08-16
> >
> > Dear Reviewer,
> >
> > As we near the midpoint of our discussion period, we would like to confirm whether we have successfully addressed the raised concerns in your review. Should any lingering issues require further attention, please let us know at your earliest convenience. This will enable us to address them promptly.
> >
> > We appreciate your time and effort in enhancing the quality of our manuscript.
> >
> > Thank you.

---

> ### Comment · Senior_Area_Chairs · 2023-08-19
> **Please comment on the author rebutttal**
>
> Dear reviewer fh35,
> Could you please let us know your reaction to this author response.
> Thanks,
> The Senior AC for this paper.

---

> > ### Comment · Reviewer_fh35 · 2023-08-21
> >
> > Thank you for your response, I am satisfied by the answers to my and other reviewers' concerns. I will thus increase my score.

---

> > > ### Author Response · Authors · 2023-08-21
> > >
> > > Thank you for your constructive feedback, and we're pleased to hear that our responses have addressed your concerns. However, we haven't noticed the score update as mentioned in your comment. Could you kindly adjust your original review rating so it would be easier for the ACs to make the final recommendation? Thank you!

---

### Official Review · Reviewer_fsqe · 2023-07-06

**Soundness:** 3 good
**Presentation:** 3 good
**Contribution:** 3 good
**Rating:** 6
**Confidence:** 2

**Summary:**

This paper proposes to study the effectiveness of features extracted from Stable Diffusion for dense correspondences. The extracted features are compared to that of DINOv2 and shown to be complementary. A very simple fusion scheme is then proposed and evaluation on datasets for sparse and dense correspondences as well as on instance swapping with convincing results.

---

The rebuttal is convincing, I maintain my initial rating leaning towards acceptance.

**Strengths:**

With all the effort that has been poured recently in diffusion models, this empirical study regarding their use outside of generative nice images is refreshing, especially since it tackle a very low level problem (keypoint correspondences) that is somewhat very remote from image generation and yet surprisingly close since the model has to generate coherent local structures. The study shows that indeed, diffusion model features are very useful for such problems and thus sheds light on what one can expect from the latent space structure of SD.

**Weaknesses:**

- All the results are performed in zero-shot, which is a bit limiting. I am not familiar with the recent literature on correspondence problems, but I assume that a fine-tuning of the features (at the very least of the projection) is possible to see how much we can get from these models.
- All the experiments were made with stable diffusion 1.5. It would be interesting to test another model to see if the same results hold across architecture and training change and thus if it is a generic property of diffusion models.
- The fact that the best results are obtained using 2 really big models is a bit annoying. It means it is difficult to know if the results come from the nature of the methods employed (SSL and diffusion) or just from the sheer capacity of the models employed.

**Questions:**

- The timestep at which features are extracted is not clearly mentioned (is it t=100 at line 214, in that case over a total time length of how long?). What influence influence does it have? Are the features more meaning closer to the denoised image ?
- The swapping are impressive, but they are refined by running the diffusion process after wise. It would be great to see compositing before the diffusion process to evaluate the quality of the correspondences.

**Limitations:**

The paper discusses 2 limiations: namely the low resolution of the features and the computational budget to get features from a big diffusion model. The first one is probably the bigger concern for correspondences.

---

> ### Author Rebuttal · Authors · 2023-08-10
>
> **W1:** Effect of fine-tuning the features with the projection layer.
>
> **A:** We briefly explored a supervised adaptation by training a projection layer [1] on top of the extracted features, guided by the CLIP-style symmetric cross entropy loss with respect to corresponding keypoints. As in Table 1, this approach yields marked improvements across both SPair-71k and PF-Pascal datasets. Notably, the fused features consistently outperform individual SD or DINOv2 features, even under the supervised setting.
>
> |Technique|Method||SPair-71k|||PF-Pascal||
> |-|-|:-:|:-:|:-:|:-:|:-:|:-:|
> |||PCK@0.01|PCK@0.05|PCK@0.1|PCK@0.05|PCK@0.1|PCK@0.15|
> |NN-based|DINOv2|6.1|39.7|55.2|61.1|77.3|83.3|
> ||SD|6.8|42.8|56.2|63.2|77.7|84.3|
> ||Fuse|8.3|47.6|63.3|72.1|86.0|90.6|
> ||
> |Projection Layer|DINOv2|8.6|56.4|74.1|74.2|90.8|95.4|
> ||SD|10.0|56.0|71.1|77.4|89.7|93.9|
> ||Fuse|**10.7**|**62.5**|**78.2**|**80.9**|**93.6**|**96.9**|
> ||
> |Supervised Baselines|PWarpC-NC-Net|-|31.6|52.0|79.2|92.1|95.6|
> |(*: fine-tuned backbone)|CATs++*|-|-|*59.8*|*84.9*|*93.8*|*96.8*|
> ||IFCAT*|-|-|*64.4*|*88.0*|*94.8*|*97.9*|
>
> Table 1. Comparison between NN-based methods, fine-tuning a projection layer, and other supervised baselines.
>
> We will include these findings in the paper, which underscores the potential of our simple fusion strategy that already yields SOTA results in an unsupervised setup.
>
> ------
> **W2:** Test other SD models with varying architectures and training settings.
>
> **A:** We further tested two alternative variations of the SD model with the same architecture but different training settings, SD-1-3 and SD-2-1-base. We also explored two work-in-progress distilled SD architectures, SD-tiny and SD-small [2] (models released by Segmind Inc.), which have 45% and 65% fewer parameters than the base model.
>
> Table 2 reports each model’s PCK metric on the SPair-71k dataset. All SD base models exhibit similar performances for both individual and fused features.
> Despite their slight performance drops, the distilled SD-tiny and SD-small variants yield noticeable improvements when fused with DINOv2. We hope to expand this analysis further to other models in future work, e.g. Pixel diffusion (Imagen [3]), token-based models (Muse [4]), once they are publicly available.
>
> |Method|PCK@0.10|PCK@0.05|PCK@0.01|
> |-|:-:|:-:|:-:|
> |SD-tiny|41.07|28.67|5.21|
> |SD-small|51.05|38.33|6.28|
> |SD-1-3|55.30|42.72|**7.72**|
> |SD-1-5|**55.90**|**42.76**|7.01|
> |SD-2-1-base|54.43|41.68|7.19|
> ||
> |DINOv2-vitb14|55.15|39.66|6.12|
> |Fuse-vitb-tiny|56.96|42.35|7.27|
> |Fuse-vitb-small|60.36|45.08|7.95|
> |Fuse-vitb-1-3|**62.69**|**47.09**|8.25|
> |Fuse-vitb-1-5|62.61|46.60|**8.47**|
> |Fuse-vitb-2-1-base|62.22|46.10|8.40|
>
> Table 2. Comparison of different variants of SD models on SPair-71k, we use explicit captioner for fair comparison.
>
> ------
> **W3:** Ablate on the capacity of the models employed.
>
> **A:** To ablate the model capacity, we explore the use of smaller, distilled versions. In particular:
>
> - We explored DINOv2-vits14, which has about 25% network parameters of the base DINOv2 model, As in Table 3, this substantially smaller variant still delivers comparable results to the base model. This suggests that while capacity plays a role, the core techniques remain effective even with a significantly smaller model.
>
> |Method|PCK@0.10|PCK@0.05|PCK@0.01|
> |-|:-:|:-:|:-:|
> |DINOv2-vits14|53.28|37.20|5.86|
> |DINOv2-vitb14|55.15|39.66|6.12|
> |Fuse-DINOv2-vits14|61.34|46.57|7.84|
> |Fuse-DINOv2-vitb14|63.28|47.61|8.32|
>
> Table 3. Comparison of DINOv2-vit small and base model on SPair-71k.
> - In Table 2, we explored the smaller SD-tiny and SD-small variants. While these models maintain similar properties as the base, they do perform worse overall (Table 2). The performance drop can be attributed to the "in progress" nature of these distilled models as well as a decrease in capacity. In general, the fusion results indicate that the DINO and SD features are complementary even when the capacity of the individual networks is changed.
>
> ------
> **Q1:** Clarity and effect of the timesteps.
>
> **A:** Our method extracts features at the time step 100 of 1000. As shown in the Table 1 of the R1-Pbqb's Q3 response, feature extraction at different time steps doesn’t significantly affect the accuracy overall. We find the time step 100 to be optimal by searching it on the validation set.
>
> ODISE finds that t=0 yields optimal results. This would be the case for semantic segmentation where a denoised image with clear object boundaries is critical for the accuracy. However, for semantic correspondence where semantic information is also important, feature maps with more structural information at a little bit earlier denoising step may help better.
>
> ------
> **Q2:** Ablate on the refinement process for swapping.
>
> **A:** Section A.5 and Table 4 in Supplementals provide quantitative comparison on the diffusion-based refinement process. The refinement step both improves the Quality score and the CLIP score but hurts the FID scores, which is possibly due to certain artifacts that amplify the discrepancy with the distribution of real images during the refinement stage.
>
> Section B.2 in Supplementals further provides qualitative examples that the refinement process visually smooths local textures from warping results.
>
> -----
>
> **References**
>
> [1] ODISE: Open-Vocabulary Panoptic Segmentation with Text-to-Image Diffusion Models, Xu et al, 2023
>
> [2] On Architectural Compression of Text-to-Image Diffusion Models, Kim et al. 2023
>
> [3] Photorealistic Text-to-Image Diffusion Models with Deep Language Understanding, Saharia et al. 2022
>
> [4] Muse: Text-To-Image Generation via Masked Generative Transformers, Chang et al. 2023

---

### Official Review · Reviewer_Pbqb · 2023-07-26

**Soundness:** 4 excellent
**Presentation:** 4 excellent
**Contribution:** 3 good
**Rating:** 7
**Confidence:** 5

**Summary:**

The paper proposes using features extracted from a stable diffusion model for dense image correspondence tasks. The paper further proposes using DINO features along with stable diffusion features for the task and empirically shows that the combination has a complementary effect--specifically SD features have good spatial localization and are smooth, whereas DINO features are sparse but accurate. Experiments are done on several datasets and the proposed feature extraction technique improves drastically over supervised, unsupervised, and weakly supervised baselines.

**Strengths:**

1. The paper is well-written, adequately inspired, and well-executed.
2. Experiments depict the importance of these large pre-trained models for the task of correspondence learning in images. Specifically, the large gains over supervised methods are quite surprising.
3. Instance swapping application clearly demonstrates the ability to do accurate correspondences between different instances of the same category of objects.

**Weaknesses:**

1. The use of pre-trained features from large models is a well-explored area of research [1,2,3]. These pre-trained models are known to perform well on downstream tasks such as semantic segmentation, detection, and classification. It is unsurprising that these features are also useful for correspondence learning.
2. Though the authors show limitations of the features. It should have been focussed more on the paper. Specifically categorizing the mistakes made by the approach and possible ways to alleviate the problems. This will be a good contribution to the community as it informs when to avoid using the proposed approach.


[1] _Unleashing text-to-image diffusion models for visual perception._ Wenliang Zhao, Yongming Rao, Zuyan Liu, Benlin Liu, Jie Zhou, and Jiwen Lu. 2023

[2] _Emerging properties in self-supervised vision transformers._ Mathilde Caron, Hugo Touvron, Ishan Misra, Hervé Jégou, Julien Mairal, Piotr Bojanowski, and Armand Joulin. 2021

[3] Open-vocabulary panoptic segmentation with text-to-image diffusion models. Jiarui Xu, Sifei Liu, Arash Vahdat, Wonmin Byeon, Xiaolong Wang, and Shalini De Mello. 2023

**Questions:**

1. Though the motivation for using these features is clear, the paper doesn't provide reasons for the complementary properties of the features from DINO and SD. It would be interesting to analyze what specifically gives rise to these complementary properties. Is it because of the training objective, architecture, datasets, etc?
2. Second and fourth rows of the right side of Figure 3 can be explained more for clarity. Specifically, if the figure also includes the original images, it will make more sense.
3. Did the authors experiment with giving different textual inputs to SD? Does including object categories in the textual categories improve the performance of correspondences?

**Limitations:**

Yes.

---

> ### Author Rebuttal · Authors · 2023-08-10
>
> **W1:** Leveraging SD / DINO features for correspondence learning.
>
> **A:** While [1,3] demonstrate that SD features are useful for depth and semantic segmentation and [2] that DINO features are useful for image retrieval and semantic segmentation, less work has focused on the tasks of semantic and dense correspondences: [4] explores using DINOv1 for these tasks but we are unaware of any published work which explores SD or DINOv2 features for correspondences.
>
> Unlike semantic segmentation or detection which are defined on a single image, **the correspondence task requires that different objects have similar representations across different images** with potentially different lighting or camera intrinsics. In addition, dense correspondence quality tends to improve when the features accurately encodes lower level details like textures, unlike segmentation or detection which benefits from features with more high-level semantics. Also different from [1], we show that SD features are surprisingly useful for correspondences in a zero-shot setting, meaning that these properties are already present in the features and do not require additional fine-tuning or processing.
>
> In addition to demonstrating that Stable Diffusion features are useful for correspondences, **our other main contribution is an in-depth analysis of the strengths and weaknesses of SD versus DINO features for correspondence**, and demonstrating that these features are surprisingly complementary. While these two features perform roughly equally, their fusion can lead to SOTA results on several correspondence benchmarks, even in a zero-shot setting with no training.
>
> ------
>
> **W2:** Limitations of the approach and potential solutions.
>
> **A:** Fig. 7 of the Supplementals show representative failures of the fusion feature, especially on matching tiny objects due to the low resolution of feature maps. We have explored different techniques for improving the resolution of the features (e.g. resizing the input or different projection layers). Our method also sometimes struggles on symmetric objects, such as Airplane image in the Pixel Warping tab in index.html, included in the Appendix. This can be overcome by exploiting geometric priors during the matching stage.
>
> Our paper primarily focuses on the unsupervised, zero shot setting at test time, which allows us to better understand the information present in the features as is. For optimal performance, we show in the rebuttal that these results can be improved through training a projection layer on top of these features (please refer to Table 1 in R2-fsqe's W1 response). We expect that even better performance can be achieved by finetuning a larger post processing network after the fusion features or just fine-tuning the existing backbone feature networks (DINOv2 and SD) in a supervised setup.
>
> We will include this discussion in the main paper.
>
> ------
>
> **Q1:** Causes of the complementary properties.
>
> **A:** Please refer to the General Response Q&A.
>
> ------
> **Q2:** Clarity on Figure 3.
>
> **A:** For the right side of Figure 3, the original image is in *the first column of the first and third rows*, and the target image is in the first column of the second and fourth rows. In the current paper, the original image has color coding to indicate the colormap for correspondences. We will also include the original image without the color coding in the updated version.
>
> ------
> **Q3:** Effect of explicit textual descriptions.
>
> **A:** The table below reports the PCK@0.10 performance of Diffusion features and Fused features in the SPair-71k 20-samples subset, when implicit and explicit textural (specifically, “a photo of {category}”) inputs are given. Overall, there are only marginal differences. The explicit textual inputs help in earlier steps (200 steps), while implicit captioner helps in denoised images. We conjecture that this is due to the implicit captioner from ODISE [1] being trained with timestep=0.
>
>
> |Method|Captioner|0|50|100|150|200|
> |-|-|:-:|:-:|:-:|:-:|:-:|
> |SD|Implicit|**54.93**|**55.67**|***56.18***|55.11|55.11|
> ||Explicit|53.58|55.63|*55.90*|**55.45**|**55.15**|
> ||||||||
> |Fuse|Implicit|**63.25**|**63.10**|***63.28***|62.46|62.50|
> ||Explicit|62.20|62.50|62.61|**62.72**|***63.32***|
>
> Table 1.  The PCK performance on SPair-71k for both implicit and explicit captioner under different timesteps. Best results between captioner are **bold**, best results among different timesteps are *italicized*.
>
> -----
> **References**
>
> [1] Unleashing text-to-image diffusion models for visual perception, Zhao et al. 2023
>
> [2] Emerging properties in self-supervised vision transformers, Caron et al. 2021
>
> [3] Open-vocabulary panoptic segmentation with text-to-image diffusion models, Xu et al. 2023
>
> [4] Deep ViT Features as Dense Visual Descriptors, Amir et al. 2021

---

> > ### Comment · Reviewer_Pbqb · 2023-08-21
> >
> > Thank you for your response. I am satisfied by the rebuttal. I have increased the score to 7.

---

> > > ### Author Response · Authors · 2023-08-21
> > >
> > > Thank you for the comments! We will further refine our paper accordingly.

---

### Author Rebuttal · Authors · 2023-08-10

## Acknowledgements

We thank the reviewers for the comments, extensive feedback, and recognition of the strengths of our work:

- **Comprehensive Analysis:** Our "extensive evaluation and analysis" of Stable Diffusion (SD) features and DINOv2 features provide "valuable insights" into their distinct properties and potential uses, as recognized by the reviewers (R3-fh35, R5-YopU).

- **Exceptional Performance:** The "impressive performance" of our method in the zero-shot setting, as well as its substantial gains over existing supervised methods, were highlighted by the reviewers (R1-Pbqb, R3-fh35, R4-NMVR, R5-YopU).

- **Practical Applications:** Our successful demonstration of instance swapping illustrated the real-world applicability of our method, as noted by the reviewers (R1-Pbqb, R3-fh35, R4-NMVR).

- **Clarity of Presentation:** The reviewers praised the "smooth logic flow" of our paper and the clarity of our explanations, noting that it was well-written and "easy to read and understand" (R1-Pbqb, R4-NMVR, R5-YopU).

- **Inspiration:** The reviewers commended our work as well-inspired, reflecting the successful combination of Stable Diffusion and DINOv2 features for semantic correspondence tasks, providing a refreshing perspective on the applications of diffusion models outside of image generation (R1-Pbqb, R2-fsqe, R3-fh35, R5-YopU).

We first address common questions by the reviewers, and then respond to individual inquiries.

------

## Questions & Responses

**Q:** (R1-Pbqb, R5-YopU) Why do Stable Diffusion and DINOv2 features behave differently?

**A:**  Due to resource limitations (e.g., time, data, and computation resources), we focus primarily on the empirical study of existing models. Unfortunately, experimentally determining the causes for feature properties often requires clean ablations with respect to data, training schemes, and architecture, and we do not have the resources to do these ablations. That said, we have several conceptual hypotheses to explain the observed differences that SD has better spatial information but worse accuracy than DINOv2:

- **Training Paradigms:** DINO's self-supervised learning training works by taking an image and producing global and local views; the global views are passed to a teacher network and used to distill information into a student's representation of both the global and local views. While this encourages "local-to-global" features [1], it also has the side effect of encouraging the features to be invariant to the training augmentations used to generate the different views (e.g., color jittering, Gaussian blur, solarization, multi-crop, and horizontal flips). **This invariance in DINO decreases the network's ability to differentiate spatial information**, e.g., the left side of an object to its right side, **particularly for symmetrical objects or objects without texture signals** (such as the bus in the bottom of Figure 3).
  **On the other hand, the SD model has been trained for text-to-image generation, which requires the model to be aware of both global object structure as well as local shape and texture cues**; for example, since the model has been trained to generate both photos and sketches of dogs, the model should be able to understand where a dog's tail is in relation to its head, even with minimal texture signal.

- **Architecture Differences:** The ViT employed by DINOv2 processes the image as a stack of patches; while the positional encoding preserves a sense of spatial awareness, attention is computed globally and less emphasis may be placed on local structure. In contrast, the convolution layers in SD’s UNet may retain more details and enhance the retention of spatial information.

Despite the lack of clean ablations, there are a few experimental data points that we can consider:
- **Model Capacity:** In this rebuttal (R2-fsqe-W2 and R2-fsqe-W3), we experiment with lower capacity versions of SD and DINOv2. The smaller distilled versions of SD have the same properties as base SD and fusing these smaller models with DINOv2 still result in improved performance on SPair-71k. The same is true of lower capacity versions of DINOv2. This suggests that capacity is not the sole reason for these features.

- **Training schemes:** We investigated models with identical architectures but diverse training protocols (R2-fsqe-W2, different variants of SD). Specifically, the SD-1-5 model is fine-tuned using different steps and datasets in comparison to SD-1-3 (195,000 steps on "laion-aesthetics v2 5+" versus 595,000 steps on "laion-improved-aesthetics"). This is also distinct from the SD-2-1-base model, which is trained on the filtered LAION-5B dataset. Despite these variations, all three models demonstrate comparable performance, indicating that these properties are robust to small variations in training schemes and datasets.

In general, we think that trying to identify specific causes and provide evidence is a very interesting question and a great direction for future work.

[1] Emerging Properties in Self-Supervised Vision Transformers, Caron et al. 2021.

---

### Decision · Program_Chairs · 2023-09-21

**Decision:**

Accept (poster)

**Comment:**

After the rebuttal and discussion phase, all reviewers recommend acceptance. The paper shows that SD features are very good for establishing dense correspondences when combined with DINO features. Before diffusion models, generative features have not been very strong for other tasks. This seems to be changing now and this paper provides insights into one area of applications: semantic correspondence and is thus valuable to the community.